# Online Optimal Tracking of Linear Systems with Adversarial Disturbances

**Farnaz Adib Yaghmaie**                                      *farnaz.adib.yaghmaie@liu.se*
*Faculty of Electrical Engineering*
*Linköping University*
*Linköping, Sweden*

**Hamidreza Modares**                                        *modaresh@msu.edu*
*The Department of Mechanical Engineering*
*Michigan State University*
*Michigan, USA*

**Reviewed on OpenReview:** *https://openreview.net/forum?id=5nVJlKgmxp*

## Abstract

This paper presents a memory-augmented control solution for the optimal reference tracking problem for linear systems subject to adversarial disturbances. We assume that the dynamics of the linear system are known and that the reference signal is generated by a linear system with unknown dynamics. Under these assumptions, finding the optimal tracking controller is formalized as an online convex optimization problem that leverages the memory of past disturbance and reference values to capture their temporal effects on the performance. That is, a (disturbance, reference)-action control policy is formalized, which selects the control actions as a linear map of the past disturbance and reference values. The online convex optimization is then formulated over the parameters of the policy on its past disturbance and reference values to optimize general convex costs. It is shown that our approach outperforms robust control methods and achieves a tight regret bound of $\mathcal{O}(\sqrt{T})$, where in our regret analysis, we have benchmarked against the best linear policy.

## 1 Introduction

Reference tracking is one of the key concepts in control theory (Isidori, 1989; Huang, 2004). In the reference tracking problem, the aim is to design a controller such that the state of the system tracks a desired reference trajectory. There are typically two common approaches for the reference tracking problem (Isidori, 1989; Huang, 2004). The first approach is called the "feedforward design". In this approach, the controller is a summation of i) a feedforward term depending on the reference signal, which is derived from the dynamics of the system and reference generator, and ii) an internal state feedback to stabilize the system in the absence of disturbances. The second approach is called the "internal model". In this approach, a dynamic controller contains an internal model of the reference signal, and the control signal is a feedback from the internal state of the controller and the state of the system. As a specific internal model approach, a proportional-integrator (PI) controller contains an integrator so it can be used for tracking a constant reference signal or rejecting a constant disturbance. Both internal model and feedforward approaches require the full knowledge of the system dynamics and reference signal generator dynamics.

The asymptotic reference following is the bare minimum requirement for the tracking control problem. To account for the transient response and the overall performance of the control design, an optimal reference tracking control problem is typically formalized and solved. One major factor that can adversely affect the performance of the tracking controllers is the presence of disturbances, which is typically ignored in the optimal reference tracking problem (Zhang et al., 2011; 2008; Huang & Liu, 2014; Kamalapurkar et al., 2015;

Adib Yaghmaie et al., 2019; Modares & Lewis, 2014; Kiumarsi & Lewis, 2015; Kiumarsi et al., 2014; 2015; 2018; Vamvoudakis et al., 2017; Carrillo & Vamvoudakis, 2020; Vamvoudakis & Lewis, 2012). In practice, dynamical systems might be affected by external disturbances originating from the environment or produced by adversaries to deteriorate the tracking performance. To account for the effect of the disturbance, it is common to assume one of the following: 1- the disturbance is generated by a dynamical system (Isidori, 1989; Huang, 2004), 2- the disturbance is Gaussian (Bertsekas, 2012), and 3- the disturbance is energy bounded and the effect of its worst-case realization is attenuated on the control performance in the robust control terminology (Doyle, 1995). In most cases, however, the disturbance is neither Gaussian nor generated by a dynamical system. Besides, the robust control approach yields conservative results because the disturbance is most likely far away from its worst-case realization (Khalil, 2002). In the related works, we discuss each case in detail.

In this paper, we design optimal tracking controllers for linear systems subject to adversarial disturbances. The adversarial disturbances are arbitrary and thus are not limited to those that are generated by a dynamical system. The reference signal to be tracked is assumed to be generated by a linear system with unknown dynamics. We assume that only the output of the reference is measurable. We design (disturbance, reference)-action control policies where a fixed-size history of disturbance and reference values are used to parameterize the proposed policy. This is partially inspired by Agarwal et al. (2019); Zhao et al. (2022), which are designed for solving the optimal regulation control problems. In contrast, we leverage the past values of both of disturbances and reference values. Using the past history of reference values is motivated by a classical result giving necessary and sufficient conditions for tracking in control theory. Indeed, our proposed policy belongs with the feedforward approach to solve the tracking problem. Our approach results in a neat parameterization of the control policy from which any general convex cost function can be optimized using online convex optimization. Indeed, we prove that the cost function is convex with respect to the parameters of the presented controller.

The resulting algorithm is online in contrast to rollout or batch-wise reinforcement learning (RL) algorithms, where it is required to collect enough samples before updating the controller (Abbasi-Yadkori et al., 2014). In sharp contrast to the robust control design approach, a history of fixed-size past disturbance and references values are calculated, stored, and used by the control policy to avoid hedging against the worst-case disturbances that rarely occur in reality (Khalil, 2002; Modares et al., 2015). We show that our proposed algorithm achieves a tight regret bound. Simulation results compare the presented approach against the $H_\infty$ control as well as the LQR control to show its superiority.

## 2 Related works

In this section, we summarize related works to the problem of optimal tracking in the presence of disturbance.

### 2.1 Tracking in the presence of disturbance

Depending on the nature of the disturbance, different strategies can be followed.

**Output regulation theory** The output regulation theory (Isidori, 1989; Huang, 2004) has been widely used to design model-free RL algorithms for solving the optimal tracking problem (Gao et al., 2017; Chen et al., 2022; Jiang et al., 2020b). It has also been leveraged to attenuate the effect of disturbances (Chen et al., 2019; Jiang et al., 2020a; Gao & Jiang, 2016; 2015). Even though RL and adaptive dynamic programming (ADP) approaches based on the output regulation theory can deal with both the optimal tracking problem and disturbance rejection, the disturbance is assumed to be generated by a dynamical system. This, however, is rarely the case in most real-world applications, which limits the applicability of the output regulation theory. Besides, ADP methods optimize a risk-neutral (expected) or risk-aware measure of the cost function under the assumption that the noise is at least i.i.d and mostly Gaussian. This is because either the value function is directly learned (policy interaction or value iteration methods) based on collected data to estimate the expected or risk-aware accumulated rewards, or the expected or risk-aware cost function or its derivative with respect to the control parameters is learned directly using data (policy gradient methods).

**Gaussian disturbance:** For linear systems with Gaussian disturbance (noise) on the system dynamics and no noise on the system's state measurements, linear quadratic regulator (LQR) control can be used to design an optimal controller for the regulation problem by minimizing a quadratic cost (Bertsekas, 2012). The feedforward gain is then calculated using the full knowledge of the dynamics of the system and reference. However, there are many control systems for which the distribution of the disturbance is not Gaussian.

**Robust control design:** For general but limited-energy disturbances, one can use the $H_\infty$-control theory to guarantee an $\mathcal{L}_2$-gain performance bound (Doyle, 1995; Khalil, 2002; Modares et al., 2015). The $H_\infty$ approach is typically overly conservative as the resulting robust controller hedges against the worst-case disturbance sequence, which rarely occurs in reality. A daunting challenge is to design non-conservative optimal tracking controllers for systems with arbitrary adversarial disturbances that do not follow assumptions such as being generated by an i.i.d. Gaussian noise sequence or by a dynamical system.

## 2.2 Notion of Optimality

To account for the transient response and the overall performance, one can introduce optimal control design to the tracking controller problem. This is usually done by designing (some part of) the controller by optimizing a performance index using the optimal control theory or reinforcement learning.

**Average or discounted costs:** In Zhang et al. (2011; 2008); Huang & Liu (2014); Kamalapurkar et al. (2015); Dierks & Jagannathan (2010) a feedforward approach is used to solve the tracking problem. The feedback part of the controller is designed by minimizing an average or discounted cost in reinforcement learning frameworks, and the feedforward part of the controller is found by dynamic inversion, assuming that the dynamics is known.

Similarly, Adib Yaghmaie et al. (2019); Modares & Lewis (2014); Kiumarsi & Lewis (2015); Kiumarsi et al. (2014; 2015) consider a feedforward approach to solve the tracking problem. But this time, both feedback and feedforward parts of the controller are designed optimally by minimizing average cost (Adib Yaghmaie et al., 2019) or discounted cost (Modares & Lewis, 2014; Kiumarsi & Lewis, 2015; Kiumarsi et al., 2014; 2015).

**Regret:** The regret compares the performance of an online control algorithm with a fixed policy in hindsight. In the context of control theory, the regret analysis is *usually* given in the regulation problem where there is no reference signal to be tracked, and the aim is to make the state vector converges to zero (Agarwal et al., 2019; Zhao et al., 2022).

In Abbasi-Yadkori et al. (2014), tracking adversarial reference trajectories with quadratic costs is considered. There are no disturbances in the problem formulation, and the regret grows as $\mathcal{O}(\log^2 N)$, where $N$ is the number of rollouts. Tracking adversarial references with convex costs is considered in Zhang et al. (2022) where an algorithm is given to estimate the state of the system based on observed data. The control signal is then generated by an algorithm, however, in control theory, the control input is usually parameterized. Examples of parameterized controllers are state feedback or disturbance-action policy (Agarwal et al., 2019) where the control input is parameterized based on the state or disturbance and the parameters are gains that are designed. Parameterization of the controller allows analysis in the context of control theory. Since the control signal is not parameterized in Zhang et al. (2022), it is difficult to perform analysis in the context of control theory. It is shown that the regret is $\mathcal{O}(\sqrt{|\mathcal{I}|})$ for time interval $\mathcal{I}$ in the time horizon $[1, T]$.

## 3 Optimal Reference Tracking Problem

*Notations and preliminaries:* Let $I$ denote an identity matrix with appropriate dimension. Let $\mathbf{1}$ and $\mathbf{0}$ denote one and zero matrices with appropriate dimensions respectively. Let $\nabla_x f$ denote the gradient of function $f(x)$ with respect to $x$. The $\mathcal{L}_2$-norm of $x$ is denoted by $\|x\|_{\mathcal{L}_2} = (\sum_{k=0}^{+\infty} \|x_k\|^2)^{\frac{1}{2}}$ where $\|x_k\|$ is the instantaneous Euclidean norm of the vector $x_k$. For matrix $A$, the spectral norm is denoted by $\|A\|$ and the Frobenius norm is denoted by $\|A\|_F$. Let $\mathbb{I}_E$ be an indicator function on set $E$. For a time-dependent variable $x_k$, the notation $x_{i:j}$, $j \geq i$ is defined as $x_{i:j} = \{x_i, x_{i+1}, .., x_j\}$. The notation $\mathcal{O}()$ is leveraged throughout the paper to express the regret upper bound as a function of $T$.

**Definition 1** *(Agarwal et al., 2019) Consider*

$$x_{k+1} = Ax_k + Bu_k$$

*and $\gamma \in [0, 1)$, $\kappa > 1$. A linear controller $K$ is $(\kappa, \gamma)$-stable if $\|K\| \leq \kappa$ and $\|\tilde{A}_K^t\|_2 \leq \kappa^2(1-\gamma)^t \; \forall t \geq 0$ where $\tilde{A}_K = A + BK$.*

### 3.1 Dynamical System and Reference Signal

Consider the following linear dynamical system

$$x_{k+1} = Ax_k + Bu_k + w_k, \tag{1}$$

where $x_k \in \mathbb{R}^n$ and $u_k \in \mathbb{R}^m$ denote the state and the control input of the system, respectively. $w_k \in \mathbb{R}^n$ denotes the adversarial input, which is captured by a general (i.e., arbitrary and unknown) disturbance. The only assumption on the disturbance is that it is bounded. We can assume without loss of generality that $x_0 = \mathbf{0}$ and push the initial condition into $w_0$.

**Assumption 1 (dynamical system)** *The pair $(A, B)$ is known and stabilizable. Moreover, the system matrices are bounded, i.e., $\|A\| \leq \kappa_a$ and $\|B\| \leq \kappa_b$.*

**Assumption 2 (disturbance)** *The disturbance sequence $w_k$ is bounded, i.e., $\|w_k\| \leq \kappa_w$ for some $\kappa_w > 0$. Moreover, $w_k = \mathbf{0}$ for $k < 0$.*

Since the system dynamics are assumed to be known, at each time $k$, $w_{1:k-1}$ are known. This is because $w_{k-1} = x_k - Ax_{k-1} - Bu_{k-1}$ and the state $x_k$ is assumed measurable.

**Remark 1** *Assumption 1 is a standard one. If the dynamics are not known, one can use Algorithm 2 in Hazan et al. (2020) and identify the dynamics by injecting random input to equation 1. In Assumption 2, we make a standard assumption that the disturbance is bounded.*

Our aim in this paper is to select the input $u_k$ such that the state of the system $x_k$ tracks an unknown linear reference signal $r_k$ generated by

$$\begin{aligned} z_{k+1} &= Sz_k, \\ r_k &= Fz_k, \end{aligned} \tag{2}$$

where $z_k \in \mathbb{R}^p$ and $r_k \in \mathbb{R}^n$ denote the state and output of the reference signal, respectively.

**Assumption 3 (reference signal)** *The following assumptions are made on the reference signal*

- *The pair $(S, F)$ is unknown, but observable.*

- *The state of the reference signal $z_k$ is not measurable but the output $r_k$ is measurable.*

- *The reference signal $r_k$ is bounded, i.e., $\|r_k\| \leq \kappa_r$.*

**Remark 2** *Even though this assumption does not cover all types of references, it can generate a large class of useful command trajectories, including unit step, sinusoidal waveforms, damped sinusoids, etc. Assumption 3 requires the reference signal to be bounded, because otherwise, the average cost defined later becomes unbounded. Therefore, Assumption 3 is usually considered when studying average cost, see for example Abbasi-Yadkori et al. (2014), Adib Yaghmaie et al. (2019). Tracking unbounded reference signals can be studied in the discounted cost settings where it is possible to guarantee the boundedness of the discounted cost by selecting the discounting factor properly, see for example Kiumarsi et al. (2014). Relaxing Assumption 3 is a direction of our future work.*

We first bring a classical result in Theorem 1 specifying the necessary and sufficient condition for the existence of a linear feedback policy to solve the classical state tracking problem, i.e., to ensure that $x_k \to r_k$, in the absence of disturbances. A linear feedback policy is defined as follows

$$u_k^{\text{lin}}(K_f) = K_{fb}x_k + K_{ff}z_k. \tag{3}$$

where $K_f = [K_{fb} \ K_{ff}] \in \mathcal{K}$ and $\mathcal{K} = \{K_f : A + BK_{fb} \text{ is } (\kappa, \gamma) - \text{stable}\}$.

The following theorem is based on Theorem 1.35 and Remark 1.36 in Huang (2004) and is modified according to the problem setup in this paper.

**Theorem 1** *(Huang, 2004)[Theorem 1.35 and Remark 1.36] Consider the dynamical system in equation 1 and the reference signal in equation 2. Assume that $w_k \equiv \mathbf{0}$, $(A, B)$ is stabilizable and $(S, F)$ is detectable. Select $K_{fb}$ such that $A + BK_{fb}$ is strongly stable. Then, the controller*

$$u_k = K_{fb}x_k + (\Gamma - K_{fb}\Pi)z_k \tag{4}$$

*solves the classical state tracking problem $x_k \to r_k$ if and only if there exist matrices $\Pi \in \mathbb{R}^{n \times p}$ and $\Gamma \in \mathbb{R}^{m \times p}$ such that*

$$\Pi S = A\Pi + B\Gamma, \quad \Pi - F = \mathbf{0}. \tag{5}$$

We show in the next lemma that even though $z_k$ is not measurable, it can be extractable from the current and past outputs of the reference if the dynamics of the reference are known.

**Lemma 1** *Assume that $(S, F)$ is observable. Let $l$ denote the observability index of equation 2; i.e., the smallest positive integer $l \geq 1$ such that*

$$\mathcal{O}_l = \begin{bmatrix} F \\ \vdots \\ FS^{l-1} \end{bmatrix} \in \mathbb{R}^{nl \times p} \tag{6}$$

*has full column rank. That is, $\text{rank}(\mathcal{O}_l) = p$. Let*

$$\begin{aligned} \mathcal{O}_l^+ &= (\mathcal{O}_l^T \mathcal{O}_l)^{-1} \mathcal{O}_l^T, \\ N &= \begin{bmatrix} N^{[1]} & \dots & N^{[l]} \end{bmatrix} = S^{l-1} \mathcal{O}_l^+, \\ N^{[s]} &\in \mathbb{R}^{p \times n}, s = 1, ..., l. \end{aligned} \tag{7}$$

*Then, the state of the reference signal can be expressed as a linear function of the current and $l - 1$ past outputs of the reference*

$$z_k = \sum_{q=0}^{l-1} N^{[l-q]} r_{k-q}. \tag{8}$$

*Proof:* See Appendix A.

The following corollary uses the results of this lemma to formalize the controller as a memory-augmented controller, which depends on the past values of the reference outputs.

**Corollary 1** *Consider the dynamical system in equation 1, the reference signal in equation 2 and $w_k \equiv \mathbf{0}$. Assume that there exist matrices $\Pi \in \mathbb{R}^{n \times p}$ and $\Gamma \in \mathbb{R}^{m \times p}$ such that equation 5 holds. Select $K_{fb}$ such that $A + BK_{fb}$ is strongly stable. Then*

$$u_k^{lin} = K_{fb}x_k + \sum_{s=0}^{l-1} (\Gamma - K_{fb}\Pi) N^{[l-s]} r_{k-s} \tag{9}$$

*solves the classical state-tracking problem $x_k \to r_k$, where $l$ is the observability index of equation 2 and $N$ is given in equation 7.*

*Proof:* The proof is based on Lemma 1 and Theorem 1.

The controller in equation 9 only guarantees asymptotic convergence of the system's state to the reference trajectory. To account for the performance, an optimal state tracking controller is typically designed by optimizing a cost function with respect to the control gains. However, the controller in equation 4 (equation 9) requires the knowledge of $\Pi$ ($\Pi$ and $N$), which is found by solving equation 5 (equation 5 and equation 7), which in turns requires the complete knowledge of the reference dynamics. As stated in Assumption 3, this knowledge is typically not available. Besides, the disturbance is either ignored in this control design approach or attenuated using overly-conservative robust control design methods. Therefore, to account for the unknown dynamics of the reference generator and to design non-conservative controllers against adversarial disturbances, a new controller is designed in the subsequent sections that leverages the past disturbances and reference values to capture their temporal effects on the performance.

## 3.2 Optimal (Disturbance,Reference)-Action Policy Design

The overall objective of this paper is to design a control policy $\pi : (x_{1:k}, w_{1:k-1}, r_{1:k}) \to u_k$ that optimizes an average cost function that captures the intention of the designer. The average cost associated with a policy $\pi$ is defined as follows

$$J_T(\pi) = \frac{1}{T} \sum_{k=1}^{T} c_k(e_k, u_k), \tag{10}$$

where $c_k$ is the rolling cost, and

$$e_k = x_k - r_k. \tag{11}$$

is the state tracking error.

**Assumption 4 (cost function)** *The cost $c_k(e_k, u_k)$ is convex in $e_k$, $u_k$. Moreover, when $\|e\|, \|u\| \leq D$, it holds that $|c_k(e_k, u_k)| \leq \beta D^2$ and $\|\nabla_e c_k(e, u)\|, \|\nabla_u c_k(e, u)\| \leq G_c D$ for some $\beta > 0$ and $G_c > 0$.*

Assumption 4 limits the cost function to general convex functions, which is more general than typical quadratic cost functions. To optimize this cost function, the following parameterization of the control policy is leveraged.

**Definition 2** *(Memory-augmented Control Policy). A (disturbance-reference)-action policy $\pi(K, M, P)$ with memory is specified by parameters $M = [M^{[0]}, ..., M^{[m_w-1]}]$, $P = [P^{[0]}, ..., P^{[m_r-1]}]$, and a fixed matrix $K$. At every time $k$, this policy chooses the action $u_k$ at a state $x_k$ using the following parameterized controller*

$$u_k^\pi(K, M, P) = K x_k + \sum_{t=1}^{m_w} M^{[t-1]} w_{k-t} + \sum_{s=0}^{m_r-1} P^{[s]} r_{k-s}, \tag{12}$$

*Since the policy parameters will be learned, and thus are changing over time, we refer to $M_k = [M_k^{[0]}, ..., M_k^{[m_w-1]}]$ and $P_k = [P_k^{[0]}, ..., P_k^{[m_r-1]}]$ as the policy parameters at time $k$.*

We call the controller $u_k^\pi$ in equation 12 a *memory-augmented control policy* which is linear in the state $x_k$, the history of the reference signal of length $m_r$ and the history of disturbance of length $m_w$.

Similar to Agarwal et al. (2019); Zhao et al. (2022), we make the following assumption on the control parameters $M$, $P$. This assumption is a prerequisite for proving the theoretical results related to our algorithm. It is enforced by using projected gradient descent in the algorithm.

**Assumption 5** *The control policy $\pi(K, M, P)$ in Definition 2 satisfies*

- *The control gain $K$ makes $A + BK$ $(\kappa, \gamma)$-strongly stable.*

- *The control parameters $Y := [M, P]$ satisfy $Y \in \mathcal{Y}$ with domain $\mathcal{Y} = \{Y = [M^{[0]}, .., M^{[m_w-1]}, P^{[0]}, ..., P^{[m_r-1]}] \mid \|M^{[t]}\|, \|P^{[t]}\| \leq \kappa_b \kappa^3 (1-\gamma)^t\}$.*

**Remark 3** *In the absence of the reference signal $r_k \equiv \boldsymbol{0}$, $\forall k$, the controller in equation 12 is simplified to the disturbance-action policy in Agarwal et al. (2019). The optimal reference tracking problem leads to a challenge of designing the controllers parameters such that the output regulator equations in equation 5 are implicitly solved by solving the optimal control problem. Note that one does not need to know l to design the controller. Indeed one can select $m_r$ large enough. In Theorem 4, we specify how to select $m_w$, $m_r$.*

*Problem 1* (Optimal Tracking Against Adversarial Disturbances): Consider the system in equation 1 under Assumptions 1 and 2. Let the reference signal be generated by equation 2 under Assumption 3. Design an algorithm or control policy that generates the control actions in the form of equation 12 to optimize the convex cost function in equation 10.

As we discuss in the sequel, we aim to propose an algorithm to learn the parameters $M$ and $P$ in equation 12 to achieve optimality. Even though other control techniques, such as Lyapunov reconstruction and sliding mode control (Khalil, 2002) can be leveraged to design robust tracking controllers, they typically only concern the stability of the system and do not provide guaranteed performance with respect to a given cost.

## 4 Properties of Memory-augmented Control Policies

To solve Problem 1, we select the controller gain $K$ in the controller in equation 12 to stabilize the dynamics in the absence of disturbance $w_k$ and reference $r_k$. We keep $K$ unchanged and aim to learn $M$ and $P$ to achieve optimality. The learning procedure is given and discussed in detail in Section 5. Before presenting the learning algorithm and proving its regret analysis, the following results are needed.

For the standard linear controller in the form of equation 3, in the presence of an adversarial or arbitrary disturbance, the $H_\infty$ control design finds the gains $K_{fb}$, $K_{ff}$ to attenuate the effect of the disturbance on the cost function. However, besides its conservativeness, as shown next, the cost function $c_k(e_k, u_k)$ is not convex in $K_{fb}$, $K_{ff}$, which makes the online control design intractable. To circumvent this difficulty and to avoid the design of an overly-conservative controller, a controller in the form of equation 12 is designed. We show next that, first, the cost function $c_k(e_k, u_k)$ is convex in $M$, $P$ (see Lemma 2), and, second, equation 12 can approximate any linear feedback policy in the form of equation 3 (see Theorem 2). Therefore, the presented memory augmented or (disturbance, reference)-action controller is favorable over the linear feedback policy $u_k^{\text{lin}}$.

**Lemma 2** *Consider the dynamical system and reference signal in equation 1-equation 2. Then, the cost function $c_k(e_k, u_k)$ is convex in $M$, $P$ for the memory-augmented controller in the form of equation 12, but is not convex in $K_{fb}$, $K_{ff}$ for the memoryless controller in the form of equation 3.*

*Proof:* See Appendix B.

**Remark 4** *Based on Lemma 2, the cost function $c_k(e_k, u_k)$ is convex with respect to the parameters of the memory-augmented control policy in equation 12. This allows online optimization of the parameters of the memory-augmented control policy for streaming settings using gradient descent. Nevertheless, as shown in Lemma 2, the optimization problem in hand is not convex in the gains of the linear feedback policy, which makes it intractable for online learning.*

Since the policy parameters will be learned, they change over time. In this case, the following Lemma shows that the state at time $k$ depends on the entire past memory of the control parameters. Let $\tilde{A}_K = A + BK$ and define

$$\Psi_{k,y}^{K,h}(M_{k-h-1:k-1}) = \tilde{A}_K^y \mathbb{I}_{y \leq h-1} + \sum_{j=0}^{h-1} \tilde{A}_K^j BM_{k-j-1}^{[y-j-1]} \mathbb{I}_{1 \leq y-j \leq m_w}, \tag{13}$$

$$\psi_{k,z}^{K,h}(P_{k-h-1:k-1}) = \sum_{j=0}^{h-1} \tilde{A}_K^j BP_{k-j-1}^{[z-j-1]} \mathbb{I}_{1 \le z-j \le m_r}. \tag{14}$$

**Lemma 3** *Let $x_k^\pi$ be the state attained upon execution of the policy $\pi(K, M_{0:k-1}, P_{0:k-1})$ that generates the control input in equation 12 at time $k$. Then, assuming $x_0 = 0$, one has*

$$
\begin{aligned}
x_k^\pi = x_k^K(M_{0:k-1}, P_{0:k-1}) = &\tilde{A}_K^h x_{k-h}^\pi + \sum_{y=0}^{m_w+h-1} \Psi_{k,y}^{K,h}(M_{k-h-1:k-1}) w_{k-y-1} \\
&+ \sum_{z=0}^{m_r+h-1} \psi_{k,z}^{K,h}(P_{k-h-1:k-1}) r_{k-z}.
\end{aligned} \tag{15}
$$

*or equivalently*

$$x_k^\pi = x_k^K(M_{0:k-1}, P_{0:k-1}) = \sum_{y=0}^{k-1} \Psi_{k,y}^{K,k}(M_{0:k-1}) w_{k-y-1} + \sum_{z=0}^{k-1} \psi_{k,z}^{K,k}(P_{0:k-1}) r_{k-z}. \tag{16}$$

*Proof:* See Appendix C.

The result of Lemma 3 shows that the memory length grows with time, which is not feasible for developing an online gradient descent-based algorithm for learning the policy parameters. Inspired by Agarwal et al. (2019), we present a truncated method that truncates the state with fixed memory lengths for both the disturbance and the reference. We also define a truncated cost accordingly.

More specifically, we truncate the state with a fixed memory length $H$. Let $\tilde{x}_k^\pi, \tilde{u}_k^\pi, f_k$ denote the truncated state, input and cost if the system would have started at $\tilde{x}_{k-H}^\pi = \mathbf{0}$. By setting $H = h$ in equation 15 and using equation 12, $\tilde{x}_k^\pi, \tilde{u}_k^\pi$ read

$$\tilde{x}_k^\pi(M_{k-H-1:k-1}, P_{k-H-1:k-1}) = \tag{17}$$
$$\sum_{y=0}^{m_w+H-1} \Psi_{k,y}^{K,H}(M_{k-H-1:k-1}) w_{k-y-1} + \sum_{z=0}^{m_r+H-1} \psi_{k,z}^{K,H}(P_{k-H-1:k-1}) r_{k-z},$$
$$\tilde{u}_k^\pi(M_{k-H-1:k}, P_{k-H-1:k}) = \tag{18}$$
$$K\tilde{x}_k^K(M_{k-H-1:k-1}, P_{k-H-1:k-1}) + \sum_{t=1}^{m_w} M_k^{[t-1]} w_{k-t} + \sum_{s=0}^{m_r-1} P_k^{[s]} r_{k-s},$$

and the truncated cost $f_k$ reads

$$
\begin{aligned}
f_k(M_{k-H-1}, ..., M_{k-1}, &P_{k-H-1}, ..., P_{k-1}) \\
&= c_k(\tilde{x}_k^\pi(M_{k-H-1:k-1}, P_{k-H-1:k-1}) - r_k, \tilde{u}_k^\pi(M_{k-H-1:k}, P_{k-H-1:k})).
\end{aligned} \tag{19}
$$

**Remark 5** *To compute the truncated state, input and cost numerically, one starts from $\tilde{x}_{k-H}^\pi = \mathbf{0}$ and steps the following dynamics*

$$\tilde{x}_{t+1}^\pi = A\tilde{x}_t^\pi + B\tilde{u}_t^\pi + w_t.$$

*for $H$ steps to get $\tilde{x}_k^\pi$. At each $t \in [k-H, k-1]$, $\tilde{u}_t^\pi$ is calculated from equation 18 using the past values of the disturbance and reference. Then, the truncated cost is calculated by using $\tilde{x}_k^\pi, \tilde{u}_k^\pi$ in $c_k$.*

In Appendix D, we bring several lemmas which will be used to prove the main results in Theorems 2-4. Specifically,

- Lemma 4 gives bounds for $\Psi_{k,y}^{K,h}, \psi_{k,z}^{K,h}$ in equation 13-equation 14.

- Lemma 5 gives bounds on the states and inputs.

- Lemma 6 gives the tracking error bound.

- Lemma 7 defines the Lipschitz condition on the truncated cost.

- Lemma 8 gives a bound on the gradient of the truncated cost.

**Theorem 2** *Consider the dynamical system and reference signal in equation 1-equation 2. Let Assumptions 1-4 hold. Let $l$ denote the observability index of equation 2. Let $u_k^{lin}$ in equation 3 be a linear feedback policy with $K_{fb}$ being $(\kappa, \gamma)$-strongly stable. Then, for any $(\kappa, \gamma)$-strongly stable $K$, there exists a memory-augmented policy of form equation 12, with $l \leq m_r$ and*

$$
\begin{aligned}
M^{[t]} =& (K_{fb} - K)(A + BK_{fb})^t, \quad 0 \leq t < m_w \\
P^{[0]} =& K_{ff} N^{[l]}, \\
P^{[s]} =& \sum_{q=0}^{\min(s-1, l-1)} (K_{fb} - K)(A + BK_{fb})^{s-q-1} BK_{ff} N^{[l-q]} \\
& + \mathbb{I}_{0<s<l} K_{ff} N^{[l-s]}, \qquad\qquad 0 < s < m_r,
\end{aligned}
\tag{20}
$$

*such that $u_k^\pi$ in equation 12 approximates $u_k^{lin}$ in equation 3. Let Assumption 5 hold. For $k > \max(m_w, m_r)$*

$$
\|u_k^{lin} - u_k^\pi\| \leq \gamma^{-1} \kappa_b \kappa^3 (1-\gamma)^{m_w} \kappa_w + \gamma^{-1} \kappa_b \kappa^3 (1-\gamma)^{m_r} \kappa_r.
$$

*Proof.* See Appendix E.

**Remark 6** *In Theorem 2, we proved that equation 12 can approximate equation 3. According to equation 2, since $\|u_k^{lin} - u_k^\pi\|$ is a function of $(1-\gamma)^{m_w}$ and $(1-\gamma)^{m_r}$, the approximation error decreases for longer history lengths $m_w$, $m_r$.*

We have seen in Theorem 2 that a memory-augmented policy $u_k^\pi$ can approximate the linear feedback policy in equation 3. In the next theorem, we show that if we use $u_k^{lin}$ and $u_k^\pi$ for the same system in equation 1 with the same sequences of disturbance, the corresponding trajectories $x_k^{lin}$ and $x_k^\pi$ are close.

**Theorem 3** *Consider the dynamical system and reference signal in equation 1-equation 2. Let Assumptions 1-5 hold. Let $l$ denote the observability index of equation 2. Assume that $K_{fb}$ and $K$ are $(\kappa, \gamma)$-strongly stable. Let $x_k^\pi$ denote the trajectory of the system using the memory-augmented control policy $u_k^\pi$ in equation 12 with the parameters in equation 20 and $x_k^{lin}$ denote the trajectory of the system using the linear policy $u_k^{lin}$ in equation 3. Assume that $w_{k-i}$, $r_{k-i}$, $i < k$ are the same in both cases. Then $x_k^\pi$ is close to $x_k^{lin}$. More specifically for $k > \max(m_w, m_r)$*

$$
\|x_k^{lin} - x_k^\pi\| \leq \gamma^{-2} \kappa_b^2 \kappa^5 (1-\gamma)^{m_w} \kappa_w + \gamma^{-2} \kappa_b^2 \kappa^5 (1-\gamma)^{m_r} \kappa_r.
$$

*Proof.* See Appendix F.

## 5 Memory-augmented online state-tracking algorithm

In this section, we will give an algorithm to tune the parameters of the linear memory-augmented policy $u_k^\pi$ in equation 12, namely $M$, $P$ to provide optimality in terms of minimization of the average cost in equation 10. Note that we consider optimizing over the class of memory-augmented policies $u_k^\pi$ in equation 12 *not* the class of linear feedback policy $u_k^{lin}$ in equation 3. The reason is that $c_k$ is convex with respect to $M$, $P$ appearing in $u_k^\pi$ but is not convex in $K_{fb}$, $K_{ff}$ in $u_k^{lin}$, see Lemma 2. The following Algorithm 1 optimizes the truncated cost $f_k$ using the gradient descent method.

### 5.1 The memory-augmented algorithm

Algorithm 1 summarizes the online state tracking procedure. In **Line 1**, the algorithm is initialized by selecting a stabilizing controller gain $K$ and setting $M$, $P$ arbitrarily. One way to select $K$ is by solving a Linear Quadratic Regulator (LQR) problem. After initiating the algorithm, the online procedure starts. In **Line 3**, the current output of the reference signal $r_k$ is recorded. Then, $u_k^\pi$ in equation 12 is calculated and applied to the system. Then, in **Line 4**, the next state of the system $x_{k+1}$ is observed and the disturbance $w_k$ is recorded. In **Line 5**, the algorithm suffers the cost $c_k(e_k, u_k)$. Then, the truncated state and inputs are calculated from equation 17-equation 18 using the latest values of $M$, $P$ and the truncated cost $f_k(M, ..., M, P, ..., P)$ is computed from equation 19. In **Line 6**, the weights $M$, $P$ are updated using projected gradient descent on the truncated cost $f_k(M, ..., M, P, ..., P)$, see equation 21 where $\Pi_M$, $\Pi_P$ denote projection onto the set of matrices with appropriate dimensions and bounded norms as specified in Assumption 5, and $\eta$ is the learning rate.

---

**Algorithm 1** Online state tracking algorithm

---

1: **Initialize:** Select a stabilizing $K$ and set $M$, $P$ arbitrarily.
2: **for** $k = 1, .., T$ **do**
3:     Record $r_k$ and execute $u_k^\pi$ in equation 12.
4:     Observe $x_{k+1}$ and record $w_k = x_{k+1} - Ax_k - Bu_k$.
5:     Suffer $c_k(e_k, u_k)$. Compute $f_k(M, ..., M, P, ..., P)$ in equation 17-equation 19.
6:     Update $M$, $P$

$$M = \Pi_M(M - \eta\nabla_M f_k(M, ..., M, P, ..., P)),$$
$$P = \Pi_P(P - \eta\nabla_P f_k(M, ..., M, P, ..., P)). \tag{21}$$

---

**Remark 7** *Algorithm 1 is online: it updates $M$, $P$ in each time step to minimize $f_k(M, ..., M, P, ..., P) = c_k(\tilde{x}_k^\pi(M, ..., M, P, ..., P) - r_k, \tilde{u}_k^\pi(M, ..., M, P, ..., P))$. Since $u_k^\pi$ approximates $u_k^{lin}$ (see Theorem 2), Algorithm 1 tries to approximate a linear feedback policy minimizing the cost $c_k$. There are two important characteristics for Algorithm 1. 1) The cost function $c_k$ does not need to be quadratic but convex. Note that in the classical approaches the cost is quadratic (Bertsekas, 2012; Khalil, 2002). 2) Parameterization of the controller $u_k^\pi$ based on the recent values of $w_k$ and $r_k$, and online tuning help us to approximate the best linear feedback policy for the recent values of $w_k$ and $r_k$. Clearly, it is less conservative than selecting the linear feedback policy for the worst case disturbance in the $H_\infty$ method. This results in a lower average cost; see the simulation results in Section 6.*

Note that at each time step $k$, $w_{1:k-1}, r_{1:k}$ are known (see **Lines 3-4** in Algorithm 1) and $w_k, r_k \equiv \mathbf{0}$ for $k < 0$ (see Assumptions 3 and 2). As such, equation 17-equation 19 are computable.

### 5.2 Regret Analysis

The standard measure for online control based on the gradient descent is the policy regret (Agarwal et al., 2019), which is defined here as the difference between the cumulative cost of the designed parameterized control policy $\pi$ learned by Algorithm 1 and that of the optimal linear control policy in the form of equation 3. The reason for selecting the class of linear policies as the baseline is twofold. Firstly, since we do not have any information about the disturbance, we compare it against the best policy when there is no disturbance in the system which is the class of linear policies. Secondly, we select the class of linear policies so that the theoretical analysis of the regret is possible.

**Definition 3** *Consider the system in equation 1. Let the control policy be deigned to generate the control action $u_k$ in equation 12 at time $k$. Let Algorithm 1 be used to update the parameters of $u_k$. Then, its regret*

*is defined as*

$$Regret = \sum_{k=1}^{T} c_k(e_k, u_k) - \min_{K_f \in \mathcal{K}} T J_T(K_f) \tag{22}$$

*where $J_T(K_f)$ is the average cost in equation 10 of the linear feedback controller in equation 3.*

The regret compares the performance of Algorithm 1 generating controllers from the class of feasible memory-augmented control policies with the best linear control policy in hindsight.

**Theorem 4** *Suppose Algorithm 1 is executed under Assumptions 1-5. Let $H = m_w = m_r$. For a fixed $T$, select the learning rate $\eta$ and the memory size $H$ to satisfy $\eta = \mathcal{O}(\frac{1}{\sqrt{T}})$ and $H = \mathcal{O}(\log T)$ to solve Problem 1. Then,*

$$Regret = \mathcal{O}(\sqrt{T}) \tag{23}$$

*Proof:* See Appendix G.

According to Theorem 4, the average cost; i.e. $\frac{1}{T}\sum_{k=1}^{T} c_k(e_k, u_k)$ is always bounded when the parameters are learned

$$\frac{1}{T}\sum_{k=1}^{T} c_k(e_k, u_k) \le \frac{1}{T} \min_{K_f \in \mathcal{K}} \sum_{k=1}^{T} c_k(e_k, u_k) + \mathcal{O}(\frac{1}{\sqrt{T}}). \tag{24}$$

The above inequality gives an upper bound for the average cost by Algorithm 1.

## 6 Simulation results

In this section, we give our simulation results.

### 6.1 The dynamical system, reference and cost function

We consider the dynamical system as

$$x_{k+1} = \begin{bmatrix} 1 & 1 \\ 0 & 1 \end{bmatrix} x_k + \begin{bmatrix} 1 & 0 \\ 0 & 1 \end{bmatrix} u_k + w_k, \tag{25}$$

and the reference signal is generated by

$$z_{k+1} = \begin{bmatrix} 0 & 1 & 0 \\ -1 & 1.5 & 0 \\ 0 & 0 & 1 \end{bmatrix} z_k, \; z_0 = [1, -2, 0.5]^T,$$
$$r_k = \begin{bmatrix} 1 & 0 & 0 \\ 0 & 0 & 1 \end{bmatrix} z_k. \tag{26}$$

where

$$x_k = \begin{bmatrix} x_{1k} \\ x_{2k} \end{bmatrix}, \; w_k = \begin{bmatrix} w_{1k} \\ w_{2k} \end{bmatrix}, \; r_k = \begin{bmatrix} r_{1k} \\ r_{2k} \end{bmatrix}, \; e_k = \begin{bmatrix} e_{1k} \\ e_{2k} \end{bmatrix} = \begin{bmatrix} x_{1k} - r_{1k} \\ x_{2k} - r_{2k} \end{bmatrix}.$$

We consider a quadratic cost with $Q = 20I_2$, $R = I_2$; that is

$$c_k = e_k^T Q e_k + u_k^T R u_k.$$

Note that the algorithm can handle any convex cost function. The choice of a quadratic cost is to enable comparison with the classical control approaches like Linear Quadratic Regulator (LQR) and $H_\infty$ controllers.

### 6.2 Disturbances

For the simulation, we consider 6 cases for the disturbance. In each case, we generate the disturbance in the beginning of the simulation so the disturbance sequence is the same for all algorithms. In the first three cases, the disturbance is randomly generated. They are useful for comparing algorithms for stochastic disturbances. For the others, we consider continuous disturbances; they are useful to study the performance of the algorithms when the disturbance is not stochastic.

- **Gaussian disturbance:** $w_{1k} \sim \mathcal{N}(0, 0.01)$, $w_{2k} \sim \mathcal{N}(0, 0.01)$. It is well known that the LQR is the optimal controller for the stabilization of the system in equation 1 (Bertsekas, 2012). Note that the support of the Gaussian noise is not finite, even though our theoretical results require the disturbance to be bounded. This bound, however, does not need to be known and could be large. This is in contrast to the robust control methods such as $H_\infty$. In our simulation results, the Gaussian noise generator (numpy.random.normal) by Numpy in Python is used which generates bounded samples.

- **Random walk disturbance:** $w_k = 0.999 w_{k-1} + \eta_{k-1}$, $\eta_{k-1} \sim \mathcal{N}(\mathbf{0}, 0.01)$. When the noise is a random walk, the optimal controller is an LQR. To see this point, we replace the random walk disturbance in equation 1

$$x_{k+1} = Ax_k + Bu_k + 0.999 w_{k-1} + \eta_{k-1}.$$

Note that in each time step $k$, the state $x_k$ is measured and according to Assumption 1, $w_{k-1}$ is known. Introducing a new state variable $\bar{x}_k = [x_k^T, w_{k-1}^T]$, we have

$$\bar{x}_{k+1} = \begin{bmatrix} A & 0.999I \\ \mathbf{0} & 0.999I \end{bmatrix} \bar{x}_k + \begin{bmatrix} B \\ \mathbf{0} \end{bmatrix} u + \begin{bmatrix} I & \mathbf{0} \\ \mathbf{0} & I \end{bmatrix} \eta_{k-1}. \tag{27}$$

Hence, equation 1 with a random walk disturbance can be seen as the extended system in equation 27 where the noise $\eta_{k-1}$ is Gaussian. As a result, the optimal controller is the LQR for the extended system in equation 27.

- **Uniformly sampled disturbance** We assume that the disturbance is uniformly sampled from the interval $[0, 1]$.

- **Constant disturbance:** $w_{1k} = w_{2k} = 1$.

- **Amplitude modulation disturbance:** $w_{1k} = w_{2k} = \sin(6\pi k/500) \sin(8\pi k/500)$.

- **Sinusoidal disturbance:** $w_{1k} = w_{2k} = \sin(8\pi k/100)$.

### 6.3 The compared control approaches

We compare our online tracking algorithm with some other linear control classes such as the $LQR$ and $H_\infty$ control approaches: both of them optimize a quadratic performance index and they are also optimal for the Gaussian and worst-case disturbances. As such, they present the best possible performance for an algorithm for example when the disturbance is Gaussian or worst-case. We would like to remind the reader that LQR and Linear Quadratic Gaussian (LQG) controller are sometimes used interchangeably when the system's states are assumed measurable. We use the terminology of LQR in this paper. We also compare our online tracking algorithm with a combination of the online control algorithm in (Agarwal et al., 2019) and the feedforward approach. Other adaptive algorithms which do not optimize a performance index or do not consider disturbance in the dynamics are not included.

In our simulation results, we study the following algorithms:

- **Online state tracking in Algorithm 1:** Let $P_r$ be the solution to the Algebraic Riccati Equation $ARE(A, B, Q, R)$. We select $K$ in equation 12 as

$$K = -(R + B^T P_r B)^{-1} B^T P_r A. \tag{28}$$

We keep $K$ unchanged during running the algorithm. We set $H = 5$, $m_r = 5$, $m_w = 5$, $\eta = 0.0001$ and initialize $M = \mathbf{0}$, $P = \mathbf{0}$ . We do not use any information about the dynamics of the reference signal; we only use measured outputs of the reference signal $r_k$ in this algorithm. We also do not use any information about the disturbance in this algorithm.

- **Online control algorithm with a fixed feedforward gain:** It is possible to combine the online control algorithm in (Agarwal et al., 2019) with a feedforward approach. We call this approach *Online control algorithm with a fixed feedforward gain*:

  1. Learn an observable canonical realization of the reference generator using measured data.
  2. Select $K$ to stabilize the system's dynamics and compute $K_{ff} = \Gamma - KF$ offline. Keep $K$, $K_{ff}$ unchanged during running the algorithm.
  3. Run the Online control algorithm in (Agarwal et al., 2019) and learn $M$. Equivalently, one can run Algorithm 1, by replacing $u_k^\pi$ in line 3 with

$$u_k^\pi = Kx_k + \sum_{t=1}^{m_w} M^{[t-1]}w_{k-t} + K_{ff}z_k \tag{29}$$

  and skipping updating $P$ in line 6.

The initial motivation for considering equation 29 is to decouple the feedback and feedforward design, to design the feedforward gain offline and the feedback gains online. Let $e_k = x_k - r_k$. Using equation 5

$$
\begin{aligned}
e_{k+1} = x_{k+1} - r_{k+1} &= Ax_k + BKx_k + B\sum_{t=1}^{m_w} M^{[t-1]}w_{k-t} + B(\Gamma - KF)z_k - FSz_k \\
&= (A + BK)x_k + B\sum_{t=1}^{m_w} M^{[t-1]}w_{k-t} + B(\Gamma - KF)z_k - (AF + B\Gamma)z_k \\
&= (A + BK)(x_k - Fz_k) + B\sum_{t=1}^{m_w} M^{[t-1]}w_{k-t} = (A + BK)e_k + B\sum_{t=1}^{m_w} M^{[t-1]}w_{k-t}.
\end{aligned}
$$

For our simulation results, we select $K$ similar to Algorithm 1 $K = -(R + B^T P_r B)^{-1}B^T P_r A$ where $P_r$ is the solution to the Algebraic Riccati Equation $ARE(A, B, Q, R)$.

- **LQR and LQR for random walk:** We apply the controller in equation 4. We select $K_{fb} = -(R + B^T P_r B)^{-1}B^T P_r A$ where $P_r = ARE(A, B, Q, R)$. We assume that we *know* the dynamics of the reference in equation 26. We then compute $K_{ff}$ according to equation 4-equation 5. To apply equation 4, we need to know the state of the reference $z_k$. We use the dynamics of the reference in equation 26 and build $z_k$ from $r_k$ according to Lemma 1.Then, we apply $u_k = K_{fb}x_k + K_{ff}z_k$.

  Note that $K_{fb}$ is the optimal feedback controller gain for stabilizing the system in equation 1 with Gaussian disturbance $w_k$ and the quadratic cost using full information of the dynamics (Bertsekas, 2012).

  We saw in Subsection 6.2 that when the disturbance is a random walk, one can extend the dynamics according to equation 27. The extended dynamics has a Gaussian disturbance and as a result, LQR for the extended dynamics is the optimal controller. In this case, we call the algorithm "*LQR for random walk*".

- $H_\infty$**-control:** In the $H_\infty$, the controller is defined to have a finite $\mathcal{L}_2$-gain with respect to the worst-case disturbance. The $H_\infty$ controller is of the form in equation 4. We design $K_{fb}$ for the system in equation 1 such that $\frac{||\sqrt{Q}x||_{\mathcal{L}_2}}{||w||_{\mathcal{L}_2}} \leq 1.5$. Note that 1.5 is the best achievable $\mathcal{L}_2$-gain for this system. We assume that we *know* the dynamics of the reference in equation 26. We then compute $K_{ff}$ according to equation 4-equation 5. To apply equation 4, we need to know the state of the reference

$z_k$. We use the dynamics of the reference in equation 26 and build $z_k$ from $r_k$ according to Lemma 1.Then, we apply $u_k = K_{fb}x_k + K_{ff}z_k$.

The $H_\infty$-control results in a conservative controller as it guarantees a finite $\mathcal{L}_2$-gain for the worst-case disturbance.

**Remark 8** *The online control algorithm with a fixed feedforward gain might not return an optimal solution. Let $K_{fb}^*$ denote the optimal linear feedback gain for the problem setup. If there is no disturbance in equation 1, i.e. $w_k \equiv 0$, the associated feedforward gain, according to the tracking theory in Theorem 1, is $K_{ff}^* = \Gamma - K_{fb}^* F$. Now, assume that the disturbance is present and the online control algorithm is used to design the controller. In the online control algorithm, the parameter $M$ is learned such that $Kx_k + \sum_{t=1}^{m_w} M^{[t-1]} w_{k-t}$ in equation 29 approximates $K_{fb}^* x_k$, see Lemma 5.2 of (Agarwal et al., 2019) or Theorem 2 in this paper. However, the feedforward gain in equation 29 is $K_{ff} = \Gamma - KF$ which is clearly different from $K_{ff}^*$. It means that while the optimal linear feedback for the system equation 1 is learned, the associated feedforward does not change. This may result in a higher cost, specifically when $K$ is away from $K_{fb}^*$. Our simulation results clearly confirm this claim.*

### 6.4 Performance during learning

In this subsection, we discuss the performance of the algorithms in Subsection 6.3 for the 6 cases of disturbance in Subsection 6.2. In Table 1, we summarize the final average cost; $\frac{1}{T} \sum_{k=1}^{T} c_k(e_k, u_k)$ suffered by the algorithms for $T = 10000$. We use **bold** to refer to the algorithm with the lowest average cost in each case of the disturbance.

In Fig. 1, the evolution of the average cost $J_T = \frac{1}{T} \sum_{k=1}^{T} c_k(e_k, u_k)$ in equation 10 vs. $T$ for the 6 cases of disturbance in subsection 6.2 has been shown. Algorithm 1 learns a memory-augmented policy in form of equation 12 by minimizing the truncated cost equation 19 as discussed in Subsection 5.1. One can see that the average cost for Algorithm 1 decreases as more iterations are done.

When the disturbance is Gaussian and assuming that the dynamics of the reference signal is known, the best linear feedback policy can be found by first selecting $K_{fb} = -(R + B^T P_r B)^{-1} B^T P_r A$ where $P_r = ARE(A, B, Q, R)$ and then computing $K_{ff}$ according to equation 4-equation 5 using the dynamics of the reference. This is the LQR controller which gives the best average cost. From Fig. 1a, we can see that the average costs by the online control algorithm with a fixed feedforward gain and Algorithm 1 approach the average cost with the best linear feedback policy, without knowing the nature of the disturbance. The online control algorithm with a fixed feedforward gain has a slightly better performance. The reason is that $K = K_{fb}$ is the optimal feedback gain. As a result, the feedforward gain is selected optimally while Algorithm 1 learns the feedforward part of the controller. Note that the online control algorithm with a fixed feedforward gain and Algorithm 1 cannot get exactly the performance of the LQR because the memory-augmented policy in equation 12 is an approximation of the linear feedback policy in equation 3.

A similar discussion is also valid for the case of random-walk disturbance, see Subsection 6.2, where we showed that the optimal controller for the system when the disturbance is a random walk, is obtained by solving an LQR problem for the extended system.

When the disturbance is not Gaussian or random walk, there is no analytical way to determine the best linear feedback policy. In such cases, usually, the $H_\infty$-controller is used to design a linear feedback policy to guarantee a finite $\mathcal{L}_2$-gain for the worst-case disturbance, and as such it is conservative. Indeed, if the disturbance is not the worst-case, the $H_\infty$-controller does not have the best performance. As one can see in Table 1 and Fig. 1, Algorithm 1 has lower average costs for uniformly sampled, constant, amplitude modulation, and sinusoidal disturbances.

### 6.5 Evaluation after learning

We further evaluate the performance of the learned controllers by Algorithm 1 by using them to control the system in equation 1 for $T_{\text{eval}} = 30$ steps. The reference trajectory for the evaluation is shown in Fig. 2.

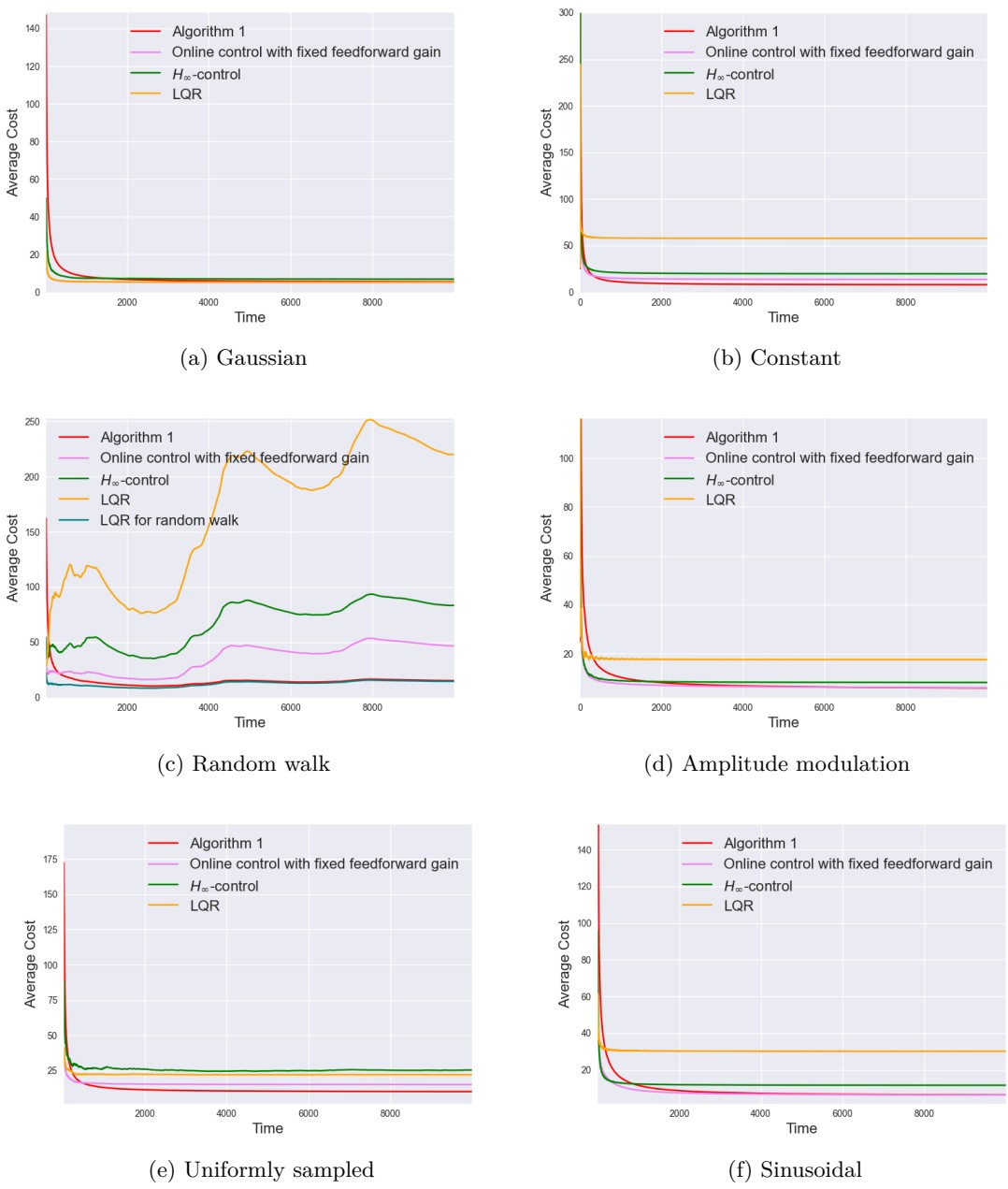

Figure 1: The evolution of the average cost $J_T = \frac{1}{T} \sum_{k=1}^{T} c_k(e_k, u_k)$ in equation 10 vs. $T$ for the presented Algorithm 1, versus, the online control algorithm with a fixed feedforward gain, the $H_\infty$ control and the LQR for Gaussian, random walk, uniformly sampled, constant, amplitude modultion and sinusoidal disturbances. In each case, we generate the disturbance in the beginning of the simulation so the disturbance sequence is the same for all algorithms. Details regarding disturbances are given in Subsection 6.2. The final values of the average costs are reported in Table 1.

Table 1: The final average costs suffered by the algorithms for running each algorithm for $T = 10000$ steps. Bold values show the lowest average cost for each case of disturbance. The algorithm LQR for random walk is the optimal controller in the case of random walk disturbance and thus it is only evaluated in this case.

| **Disturbance** | Algorithm 1 | Online control with fixed feedforward gain | LQR | $H_\infty$ | LQR for random walk |
|---|---|---|---|---|---|
| Gaussian | 5.58 | 5.42 | **5.29** | 6.93 | N.A. |
| Random walk | 15.19 | 46.67 | 219.85 | 83.26 | **14.50** |
| Uniformly sam. | **10.10** | 15.19 | 22.05 | 25.32 | N.A. |
| Constant | **8.02** | 13.78 | 57.77 | 19.75 | N.A. |
| Amplitude mod. | **5.94** | 6.17 | 17.55 | 8.21 | N.A. |
| Sinusoidal | **6.35** | 6.41 | 30.17 | 11.63 | N.A. |

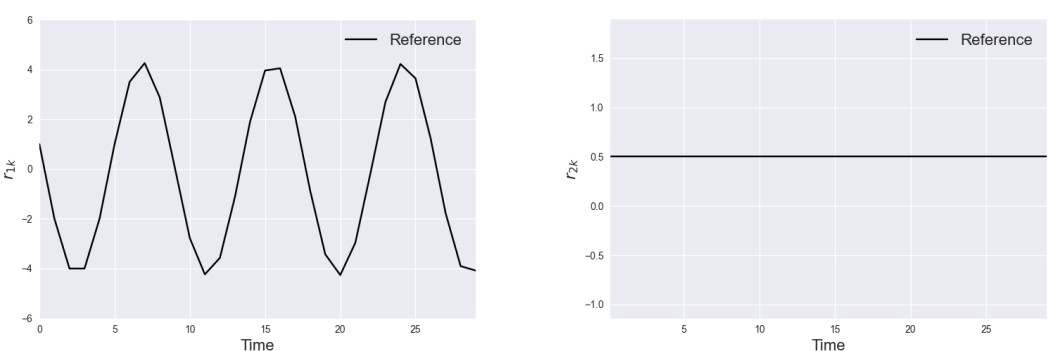

Figure 2: The reference signals used for the evaluation and comparison of different approaches under different disturbances.

We generate the disturbance in the beginning of the simulation so the disturbance sequence is the same for all algorithms during the evaluation. In Fig. 3-8, one can see the state tracking error for different cases of the disturbances using the methods in Subsection 6.3. Figures 3-8 also confirm the results in Table 1. For amplitude modulation and sinusoidal disturbances, the tracking errors by Algorithm 1 and the online control with a fixed feedforward gain are zero while other approaches have significant nonzero errors. For the constant disturbance, the tracking error by Algorithm 1 is near zero while other approaches have nonzero errors. For stochastic disturbances, it is more difficult to understand the behavior from the tracking error plots and the performance is better understood by the numbers in Table 1, but we have kept the figures for completeness of the results. For the Gaussian disturbance, the performance of Algorithm 1 is almost identical to the LQR controller which is the optimal controller in this case.

## 7 Conclusion

In this paper, we have considered the problem of state tracking in the presence of general disturbances. We have proposed a memory-augmented controller and given an online algorithm to tune the parameters of the controller. Our proposed algorithm tunes the parameters of the controller online to achieve state tracking and disturbance rejection while minimizing general convex costs. We have proved that the algorithm attains $\mathcal{O}(\sqrt{T})$-policy regret. In our future works, we will consider partially observable dynamical systems and aim to remove the bounded assumption on the reference signal.

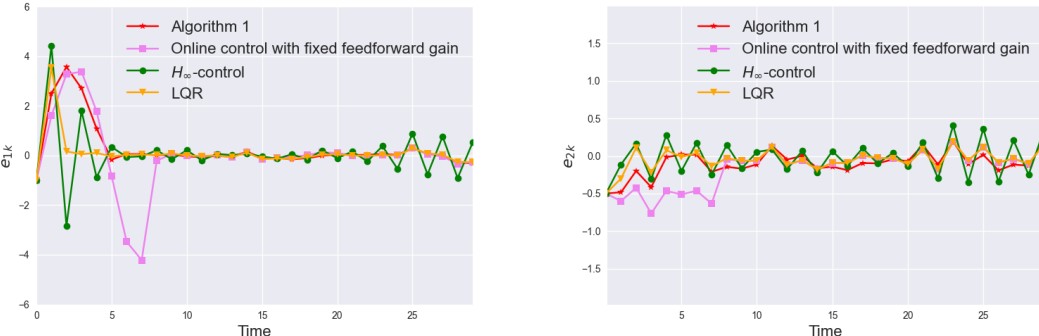

Figure 3: Tracking error for the Gaussian disturbance for the presented Algorithm 1, versus, the online control algorithm with a fixed feedforward gain, the $H_\infty$ control and the LQR control using the reference signals in Fig. 2.

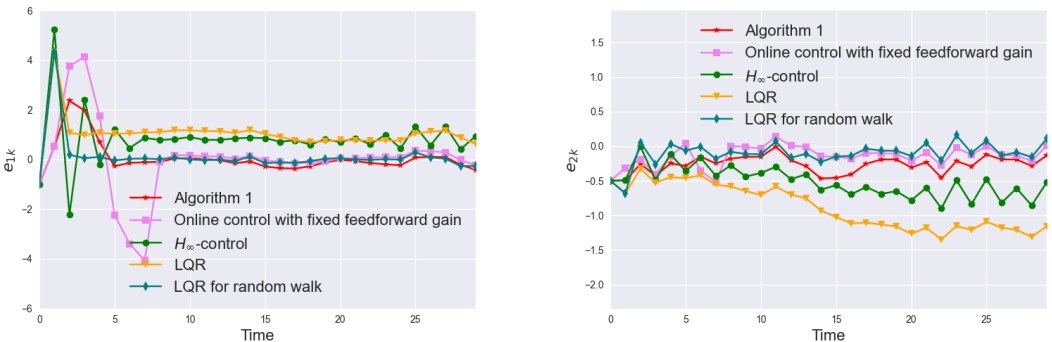

Figure 4: Tracking error for the random walk disturbance for the presented Algorithm 1, versus the online control algorithm with a fixed feedforward gain, the $H_\infty$ control, the LQR control, and the LQR for random walk using the reference signals in Fig. 2.

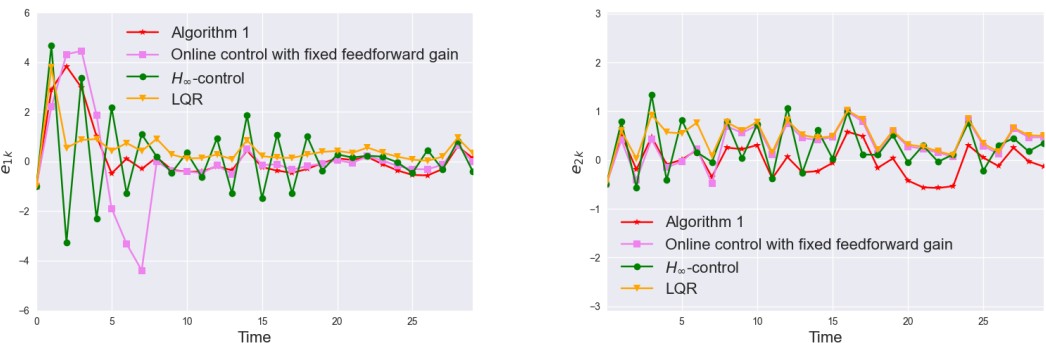

Figure 5: Tracking error for the uniformly sampled disturbance for the presented Algorithm 1, versus the online control algorithm with a fixed feedforward gain, the $H_\infty$ control and the LQR control using the reference signals in Fig. 2.

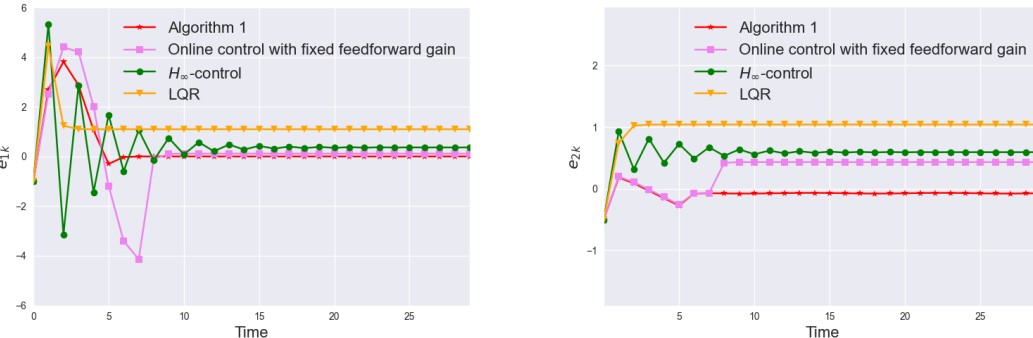

Figure 6: Tracking error for the constant disturbance for the presented Algorithm 1, versus the online control algorithm with a fixed feedforward gain, the $H_\infty$ control and the LQR control using the reference signals in Fig. 2.

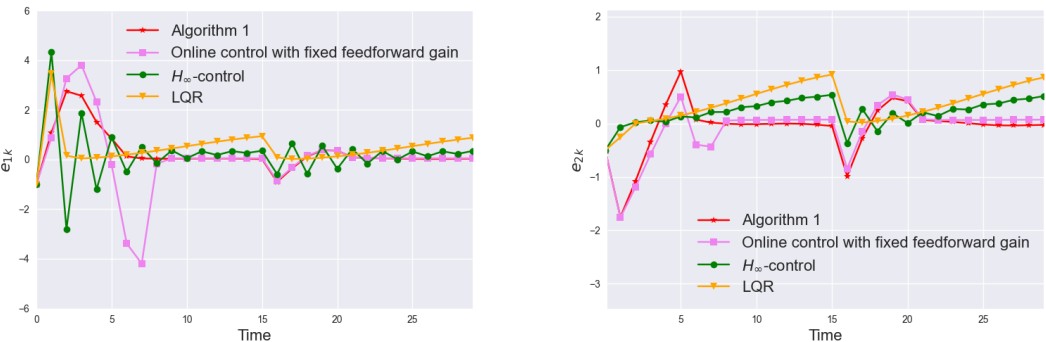

Figure 7: Tracking error for the amplitude modulation disturbance for the presented Algorithm 1, versus the online control algorithm with a fixed feedforward gain, the $H_\infty$ control and the LQR control using the reference signals in Fig. 2.

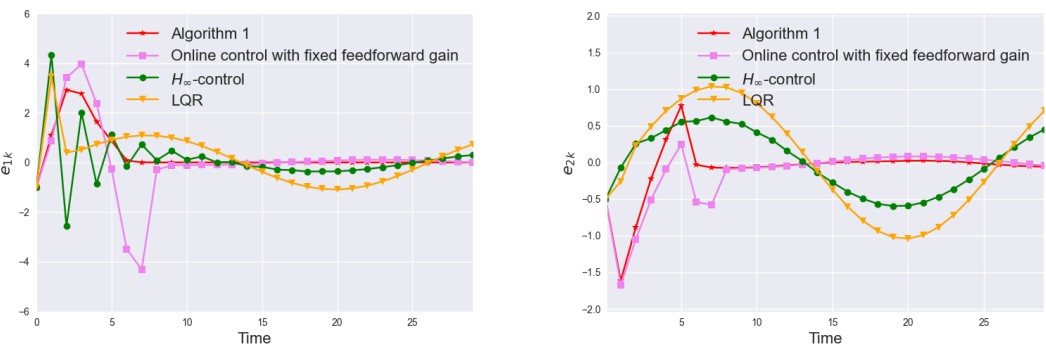

Figure 8: Tracking error for the sinusoidal disturbance for the presented Algorithm 1, versus the online control algorithm with a fixed feedforward gain, the $H_\infty$ control and the LQR control using the reference signals in Fig. 2.

**Acknowledgments**

Farnaz Adib Yaghmaie is supported by Excellence Center at Linköping–Lund in Information Technology (ELLIIT), and ZENITH. Hamidreza Modares is supported by Department of Navy award N00014-22-1-2159 issued by the Office of Naval Research.

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

# A  Proof of Lemma 1

Consider the current and the $l - 1$ past outputs of the system

$$r_{k-l+1} = Fz_{k-l+1},$$
$$z_{k-l+2} = Sz_{k-l+1},$$
$$r_{k-l+2} = Fz_{k-l+2} = FSz_{k-l+1},$$
$$\vdots$$

Continuing like this for $l$ steps, we get at the end

$$z_k = S^{l-1}z_{k-l+1},$$
$$r_k = FS^{l-1}z_{k-l+1}. \tag{30}$$

Let $\bar{r}_k = \begin{bmatrix} r_{k-l+1} \\ \vdots \\ r_k \end{bmatrix}$ be the concatenation of the outputs from $r_{k-l+1}$ to $r_k$. We have $\bar{r}_k = \mathcal{O}_l z_{k-l+1}$. Since the matrix $\mathcal{O}_l$ has full column rank $\mathrm{rank}(\mathcal{O}_l) = p$, any $p \times p$ matrix can be spanned by columns of $\mathcal{O}_l$. In particular, there exists a matrix $N$ such that $S^{l-1} = N\mathcal{O}_{l-1}$ and $N = S^{l-1}\mathcal{O}_l^+$. Using this result in the first equation in equation 30

$$z_k = S^{l-1}z_{k-l+1} = N\mathcal{O}_{l-1}z_{k-l+1} = N\bar{r}_k.$$

# B  Proof of Lemma 2

Let $x_k^{\mathrm{lin}}(K_{fb}, K_{ff})$ denote solution to the system in equation 1 using the linear feedback policy $u_k^{\mathrm{lin}}(K_{fb}, K_{ff})$ in equation 3 and $x_k^\pi(K, M, P)$ denote the solution to the system in equation 1 using the memory-augmented policy $u_k^\pi(K, M, P)$ in equation 12. We drop the arguments in $x_k^{\mathrm{lin}}$, $u_k^{\mathrm{lin}}$, $x_k^\pi$, $u_k^\pi$ for clarity in the proof. Using $u_k^{\mathrm{lin}}$, the closed-loop system reads

$$x_{k+1}^{\mathrm{lin}} = (A + BK_{fb})x_k^{\mathrm{lin}} + BK_{ff}z_k + w_k$$
$$= (A + BK_{fb})x_k^{\mathrm{lin}} + BK_{ff}\sum_{q=0}^{l-1} N^{[l-q]}r_{k-q} + w_k,$$

where we have used equation 8 in Lemma 1 to replace $z_k$ in the second line. $x_k^{\mathrm{lin}}$ reads

$$x_k^{\mathrm{lin}} = \sum_{i=1}^{k}(A + BK_{fb})^{i-1}BK_{ff}\sum_{q=0}^{l-1} N^{[l-q]}r_{k-i-q} + \sum_{i=1}^{k}(A + BK_{fb})^{i-1}w_{k-i}$$
$$= \sum_{i=1}^{k}\sum_{q=0}^{l-1}(A + BK_{fb})^{i-1}BK_{ff}N^{[l-q]}r_{k-i-q} + \sum_{i=1}^{k}(A + BK_{fb})^{i-1}w_{k-i}. \tag{31}$$

Now, we change the index in the first summation; it will simplify our derivations in other proofs. Introduce $j = i + q$ and let $1 \le j \le k$. We have two bounds for $i$ which give two bounds for $q$ after the change of the variable $j = i + q$.

- The lower bound on $i = j - q \geq 1$. From this, one should have that $q \leq j - 1$. Since $q \leq l - 1$ also in the summation, we will have that $q \leq \min(l - 1, j - 1)$.

- The upper bound on $i = j - q \leq k$. Since $j \leq k$, we have $q \geq$ (a negative number). Since $q \geq 0$ in the summation, we will have that $q \geq 0$.

Combining these two boundaries for $q$, we have $0 \leq q \leq \min(l - 1, j - 1)$. As a result, the above equation can be written as

$$x_k^{\text{lin}} = \sum_{j=1}^{k} \sum_{q=0}^{\min(l-1,j-1)} (A + BK_{fb})^{j-q-1} BK_{ff} N^{[l-q]} r_{k-j} + \sum_{i=1}^{k} (A + BK_{fb})^{i-1} w_{k-i}. \tag{32}$$

Note that $c_k$ is convex in $x_k^{\text{lin}}$, but based on equation 32, $x_k^{\text{lin}}$ is not convex in $K_{fb}$, $K_{ff}$. As a result, $c_k(e_k, u_k)$ is not convex in $K_{fb}$, $K_{ff}$ in general.

Next, we study the solution to the system in equation 1 using the memory-augmented policy $u_k^\pi$. Using $u_k^\pi$, the closed-loop system reads

$$x_{k+1}^\pi = \tilde{A}_K x_k^\pi + B \sum_{t=1}^{m_w} M^{[t-1]} w_{k-t} + w_k + B \sum_{s=0}^{m_r-1} P^{[s]} r_{k-s}.$$

As a result, $x_k^\pi$ reads

$$\begin{aligned}
x_k^\pi &= \sum_{i=1}^{k} \tilde{A}_K^{i-1} B \sum_{t=1}^{m_w} M^{[t-1]} w_{k-i-t} + \sum_{i=1}^{k} \tilde{A}_K^{i-1} w_{k-i} \\
&+ \sum_{i=1}^{k} \tilde{A}_K^{i-1} B \sum_{s=0}^{m_r-1} P^{[s]} r_{k-i-s} = \sum_{i=1}^{k} \sum_{t=1}^{m_w} \tilde{A}_K^{i-1} B M^{[t-1]} w_{k-i-t} \\
&+ \sum_{i=1}^{k} \tilde{A}_K^{i-1} w_{k-i} + \sum_{i=1}^{k} \sum_{s=0}^{m_r-1} \tilde{A}_K^{i-1} B P^{[s]} r_{k-i-s}.
\end{aligned} \tag{33}$$

Again, we change the indices in the summations. The reasoning is similar to what we discussed earlier. For the first summation, introduce $j = i + t$ and let $1 \leq j \leq k$. We have two bounds for $i$ which give two bounds for $t$ after the change of the variable $j = i + t$.

- The lower bound on $i = j - t \geq 1$. From this, one should have that $t \leq j - 1$. Since $t \leq m_w$ also in the summation, we will have that $t \leq \min(m_w, j - 1)$.

- The upper bound on $i = j - t \leq k$. Since $j \leq k$, we have $t \geq$ (a negative number). Since $t \geq 1$ in the summation, we will have that $t \geq 1$.

Combining these two boundaries for $t$, we have $1 \leq t \leq \min(m_w, j - 1)$. For the third summation, introduce $j = i + s$ and let $1 \leq j \leq k$. We can follow a similar reasoning and show that $0 \leq s \leq \min(m_r - 1, j - 1)$.

$$\begin{aligned}
x_k^\pi &= \sum_{j=1}^{k} \sum_{t=1}^{\min(m_w,j-1)} \tilde{A}_K^{j-t-1} B M^{[t-1]} w_{k-j} + \sum_{i=1}^{k} \tilde{A}_K^{i-1} w_{k-i} \\
&+ \sum_{j=1}^{k} \sum_{s=0}^{\min(m_r-1,j-1)} \tilde{A}_K^{j-s-1} B P^{[s]} r_{k-j}.
\end{aligned} \tag{34}$$

Note that $c_k$ is convex in $x_k^\pi$ and $x_k^\pi$ is linear in $M$, $P$. As a result, $c_k(e_k, u_k)$ is convex in $M$, $P$.

## C  Proof of Lemma 3

Based on Lemma 2, the expression of the state at time $k$ becomes,

$$x_k^\pi = \sum_{i=1}^{k} \sum_{t=1}^{m_w} \tilde{A}_K^{i-1} BM_{k-i}^{[t-1]} w_{k-i-t} + \sum_{i=1}^{k} \tilde{A}_K^{i-1} w_{k-i} + \sum_{i=1}^{k} \sum_{s=0}^{m_r-1} \tilde{A}_K^{i-1} BP_{k-i}^{[s]} r_{k-i-s}. \tag{35}$$

Forming $x_{k-h}^\pi$ for any $h \geq 0$, multiplying it by $\tilde{A}_K^h$, and subtracting $x_k^\pi$ from $\tilde{A}_K^h x_{k-h}^\pi$ yields

$$x_k^\pi - \tilde{A}_K^h x_{k-h}^\pi = \sum_{j=1}^{k} \tilde{A}_K^{j-1} w_{k-j} - \sum_{j=h+1}^{k} \tilde{A}_K^{j-1} w_{k-i} + \sum_{j=1}^{k} \sum_{t=1}^{m_w} \tilde{A}_K^{j-1} BM_{k-j}^{[t-1]} w_{k-j-t}$$

$$- \sum_{j=h+1}^{k} \sum_{t=1}^{m_w} \tilde{A}_K^{j-1} BM_{k-j}^{[t-1]} w_{k-j-t} + \sum_{j=1}^{k} \sum_{s=0}^{m_r-1} \tilde{A}_K^{j-1} BP_{k-j}^{[s]} r_{k-j-s} - \sum_{j=h+1}^{k} \sum_{s=0}^{m_r-1} \tilde{A}_K^{j-1} BP_{k-j}^{[s]} r_{k-j-s}$$

$$= \sum_{j=1}^{h} \tilde{A}_K^{j-1} w_{k-j} + \sum_{j=1}^{h} \sum_{t=1}^{m_w} \tilde{A}_K^{j-1} BM_{k-j}^{[t-1]} w_{k-j-t} + \sum_{j=1}^{h} \sum_{s=0}^{m_r-1} \tilde{A}_K^{j-1} BP_{k-j}^{[s]} r_{k-j-s}$$

In the second summation, introduce $y = j + t - 1$ and change the order of summations, so $y = 0, ..., h + m_w - 1$. Similarly, introduce $z = j + s$ so, $z = 0, ..., h + m_r - 1$ and then $j - 1 \to j$

$$x_k^\pi - \tilde{A}_K^h x_{k-h}^\pi = \sum_{j=0}^{h-1} \tilde{A}_K^j w_{k-j-1} + \sum_{y=0}^{m_w+h-1} \sum_{j=1}^{h} \tilde{A}_K^{j-1} BM_{k-j}^{[y-j]} w_{k-y-1} + \sum_{z=0}^{h+m_r-1} \sum_{j=1}^{h} \tilde{A}_K^{j-1} BP_{k-j}^{[z-j]} r_{k-z}$$

$$= \sum_{a=0}^{h-1} \tilde{A}_K^a w_{k-a-1} + \sum_{y=0}^{m_w+h-1} \sum_{j=0}^{h-1} \tilde{A}_K^j BM_{k-j-1}^{[y-j-1]} w_{k-y-1} + \sum_{z=0}^{h+m_r-1} \sum_{j=0}^{h-1} \tilde{A}_K^j BP_{k-j-1}^{[z-j-1]} r_{k-z}.$$

Then, based on equation 13, we have

$$x_k^\pi = x_k^K(M_{0:k-1}, P_{0:k-1}) = \tilde{A}_K^h x_{k-h}^\pi + \sum_{y=0}^{m_w+h-1} \Psi_{k,y}^{K,h}(M_{k-h-1:k-1}) w_{k-y-1}$$

$$+ \sum_{z=0}^{m_r+h-1} \psi_{k,z}^{K,h}(P_{k-h-1:k-1}) r_{k-z}.$$

Set $h = k$ and note that $y \leq k - 1$ and $z \leq k - 1$. Then, equation 16 is concluded.

## D  Supporting Lemmas

**Lemma 4** *Let Assumptions 1-5 hold. Suppose that $K$ is $(\kappa, \gamma)$-strongly stable. Then,*

$$\begin{aligned} \|\Psi_{k,y}^{K,h}\| &\leq \kappa^2 (1-\gamma)^y \mathbb{I}_{y \leq h-1} + m_w \kappa^5 \kappa_b^2 (1-\gamma)^{y-1}, \\ \|\psi_{k,z}^{K,h}\| &\leq m_r \kappa^5 \kappa_b^2 (1-\gamma)^{z-1}. \end{aligned} \tag{36}$$

*Proof:* To prove the first statement in equation 36 note that

$$\|\Psi_{k,y}^{K,h}\| \leq \|\tilde{A}_K^y \mathbb{I}_{y \leq h-1}\| + \|\sum_{j=0}^{h-1} \tilde{A}_K^j BM_{k-j-1}^{[y-j-1]} \mathbb{I}_{1 \leq y-j \leq m_w}\|$$

$$\leq \kappa^2 (1-\gamma)^y \mathbb{I}_{y \leq h-1} + \sum_{j=0}^{h-1} \kappa^2 \kappa_b^2 \kappa^3 (1-\gamma)^{y-1} \mathbb{I}_{1 \leq y-j \leq m_w}$$

$$\leq \kappa^2 (1-\gamma)^y \mathbb{I}_{y \leq h-1} + m_w \kappa^5 \kappa_b^2 (1-\gamma)^{y-1},$$

where the second inequality follows from $(\kappa, \gamma)$-stability of the controller gain $K$ and Assumption 5 which gives the bound for $M$. The proof of the second statement follows similarly.

**Lemma 5** *Let Assumptions 1-5 hold. Define*

$$
\begin{aligned}
Y_{0:k} &:= [M_{0:k}, P_{0:k}], \\
Y_{H,k} &:= [M_{k-H:k}, P_{k-H:k}].
\end{aligned}
\tag{37}
$$

$$
D := \gamma^{-1} \frac{\kappa_w \kappa^3 + (\kappa_r m_r + \kappa_w m_w)(1-\gamma)^{-1} \kappa^6 \kappa_b^2}{1 - \kappa^2 (1-\gamma)^H} + \frac{(\kappa_w + \kappa_b \kappa_n \kappa_r) \kappa_b \kappa^3}{\gamma},
$$

*where* $\kappa_n = \|K_{ff}^* \sum_{q=0}^{l-1} N^{[l-q]}\|$.

*Suppose that $K$ and $K_{fb}^*$ are $(\kappa, \gamma)$-strongly stable. Define $x_k^{lin}(K_{fb}^*, K_{ff}^*)$ as the system state corresponding to an optimal linear feedback controller. Then, one has*

$$
\max(\|x_k^\pi(Y_{0:k-1})\|, \|\tilde{x}_k^\pi(Y_{H,k-1})\|, \|x_k^{lin}(K_{fb}^*, K_{ff}^*)\|) \leq D, \tag{38}
$$

$$
\max(\|u_k^\pi(Y_{0:k})\|, \|\tilde{u}_k^\pi(Y_{H,k})\|) \leq D, \tag{39}
$$

$$
\|x_k^\pi(Y_{0:k-1}) - \tilde{x}_k^\pi(Y_{H,k-1})\| \leq \kappa^2 (1-\gamma)^H D, \tag{40}
$$

$$
\|u_k^\pi(Y_{0:k}) - \tilde{u}_k^\pi(Y_{H,k})\| \leq \kappa^3 (1-\gamma)^H D. \tag{41}
$$

*Proof:* Using equation 15, we have

$$
\begin{aligned}
\|x_k^\pi\| &\leq \|\tilde{A}_K^H\| \|x_{k-H}^\pi\| + \kappa_w \sum_{y=0}^{m_w+H-1} \|\Psi_{k,y}^{K,H}(M_{k-H-1:k-1})\| + \kappa_r \sum_{z=0}^{m_r+H-1} \|\psi_{k,z}^{K,H}(P_{k-H-1:k-1})\| \\
&\leq \kappa^2 (1-\gamma)^H \|x_{k-H}^\pi\| + \kappa_w \gamma^{-1}(\kappa^2 + m_w \kappa^5 \kappa_b^2 (1-\gamma)^{-1}) + \kappa_r \gamma^{-1}(m_r \kappa^5 \kappa_b^2)(1-\gamma)^{-1}.
\end{aligned}
$$

The above recursion satisfies

$$
\|x_k^\pi\| \leq \gamma^{-1} \frac{\kappa_w \kappa^2 + (\kappa_r m_r + \kappa_w m_w)(1-\gamma)^{-1} \kappa^5 \kappa_b^2}{1 - \kappa^2 (1-\gamma)^H}.
$$

Similarly, from equation 17, one has

$$
\begin{aligned}
\|\tilde{x}_k^\pi(Y_{H,k-1})\| &\leq \sum_{y=0}^{m_w+H-1} \|\Psi_{k,y}^{K,H}(M_{k-H-1:k-1}) w_{k-y-1}\| + \sum_{z=0}^{m_r+H-1} \|\psi_{k,z}^{K,H}(P_{k-H-1:k-1}) r_{k-z}\| \\
&\leq \gamma^{-1} \kappa_w \kappa^2 + \gamma^{-1}(\kappa_w m_w + \kappa_r m_r) \kappa^5 \kappa_b^2 (1-\gamma)^{-1} \leq D.
\end{aligned}
$$

where the last inequality is obtained because $0 \leq 1 - \kappa^2 (1-\gamma)^H \leq 1$. Moreover,

$$
\begin{aligned}
\|x_k^{lin}(K_{fb}^*, K_{ff}^*)\| &= \|\sum_{y=0}^{k-1} \tilde{A}_{K_{fb}^*}^y w_{k-y-1} + \sum_{i=0}^{k-1} \tilde{A}_{K_{fb}^*}^i B K_{ff}^* z_{k-i}\| \\
&\leq \|\sum_{y=0}^{k-1} \tilde{A}_{K_{fb}^*}^y w_{k-y-1}\| + \|\sum_{i=0}^{k-1} \tilde{A}_{K_{fb}^*}^i B K_{ff}^* \sum_{q=0}^{l-1} N^{[l-q]} r_{k-i-q}\| \\
&\leq \gamma^{-1} \kappa^2 (\kappa_w + \kappa_b \kappa_n \kappa_r) \leq D.
\end{aligned}
$$

Beside, one has

$$\|u_k^\pi(Y_{0:k})\| = \|Kx_k^\pi(Y_{0:k-1}) + \sum_{t=1}^{m_w} M^{[t-1]}w_{k-t} + \sum_{s=0}^{m_r-1} P^{[s]}r_{k-s}\|$$

$$\leq \kappa\|x_k^\pi(Y_{0:k-1})\| + \kappa_w \sum_{t=1}^{m_w} \kappa_b\kappa^3(1-\gamma)^{(t-1)} + \kappa_r \sum_{s=0}^{m_r-1} \kappa_b\kappa^3(1-\gamma)^s$$

$$\leq \gamma^{-1}\frac{\kappa_w\kappa^3 + (\kappa_r m_r + \kappa_w m_w)(1-\gamma)^{-1}\kappa^6\kappa_b^2}{1-\kappa^2(1-\gamma)^H} + \frac{(\kappa_w + \kappa_r)\kappa_b\kappa^3}{\gamma} \leq D.$$

Similarly,

$$\|\tilde{u}_k^\pi(Y_{H,k})\| = \|K\tilde{x}_k^\pi(Y_{H,k-1}) + \sum_{t=1}^{m_w} M^{[t-1]}w_{k-t} + \sum_{s=0}^{m_r-1} P^{[s]}r_{k-s}\|$$

$$\leq \kappa\|\tilde{x}_k^\pi(Y_{H,k-1})\| + \kappa_w \sum_{t=1}^{m_w} \kappa_b\kappa^3(1-\gamma)^{(t-1)} + \kappa_r \sum_{s=0}^{m_r-1} \kappa_b\kappa^3(1-\gamma)^s$$

$$\leq \gamma^{-1}\kappa_w\kappa^3 + \gamma^{-1}(\kappa_w m_w + \kappa_r m_r)\kappa^6\kappa_b^2(1-\gamma)^{-1} + \frac{(\kappa_w + \kappa_r)\kappa_b\kappa^3}{\gamma} \leq D.$$

To bound the difference between the actual and truncated state, from equation 17 and equation 15, one has

$$\|x_k^\pi(Y_{0:k-1}) - \tilde{x}_k^\pi(Y_{H,k-1})\| = \|\tilde{A}_K^H x_{k-H}^\pi(Y_{0:k-H-1})\| \leq \kappa^2(1-\gamma)^H D$$

which gives

$$\|u_k^\pi(Y_{0:k}) - \tilde{u}_k^\pi(Y_{H,k})\| \leq \|K\|\|\tilde{A}_K^H x_{k-H}^\pi(Y_{0:k-H-1})\| \leq \kappa^3(1-\gamma)^H D.$$

This completes the proof.

**Lemma 6** *Let Assumptions 1-5 hold. Suppose that $K$ is $(\kappa, \gamma)$-strongly stable. Define*

$$\kappa_z := \|\psi_{k,z}^{K,k}(P_{0:k-1}) - \Pi N^{[l-z]}\|.$$

*Then, the tracking error bound is given by*

$$\|e_k\| \leq \kappa_w\gamma^{-1}(\kappa^2 + m_w\kappa^5\kappa_b^2(1-\gamma)) + \kappa_r\gamma^{-1}(1-\gamma)^{l-1}m_r\kappa^5\kappa_b^2 + \kappa_r \sum_{z=0}^{l-1}\kappa_z. \tag{42}$$

*Proof:* First, note that $\kappa_z$ is bounded because $\psi_{k,z}^{K,k}(P_{0:k-1})$ is bounded, see Lemma 4. Using equation 5, the tracking error is defined as

$$e_k = x_k^\pi(Y_{0:k-1}) - Fz_k = x_k^\pi(Y_{0:k-1}) - \Pi z_k$$

where $x_k^\pi(Y_{0:k-1})$ is defined in equation 16. Using Lemma 1 to replace $z_k$ with a linear combination of the outputs of the reference

$$e_k = \sum_{y=0}^{k-1} \Psi_{k,y}^{K,k}(M_{0:k-1})w_{k-y-1} + \sum_{z=0}^{k-1} \psi_{k,z}^{K,k}(P_{0:k-1})r_{k-z} - \Pi \sum_{q=0}^{l-1} N^{[l-q]}r_{k-q}$$

$$= \sum_{y=0}^{k-1} \Psi_{k,y}^{K,k}(M_{0:k-1})w_{k-y-1} + \sum_{z=0}^{l-1}(\psi_{k,z}^{K,k}(P_{0:k-1}) - \Pi N^{[l-z]})r_{k-z} + \sum_{z=l}^{k-1} \psi_{k,z}^{K,k}(P_{0:k-1})r_{k-z}.$$

Using the bounds in equation 4

$$
\begin{aligned}
\|e_k\| \leq & \sum_{y=0}^{k-1}(\kappa^2(1-\gamma)^y \mathbb{I}_{y\leq k-1} + m_w\kappa^5\kappa_b^2(1-\gamma)^{y-1})\kappa_w + \sum_{z=0}^{l-1}\|\psi_{k,z}^{K,k}(P_{0:k-1}) - \Pi N^{[l-z]}\|\kappa_r \\
& + \sum_{z=l}^{k-1} m_r\kappa^5\kappa_b^2(1-\gamma)^{z-1}\kappa_r \\
\leq & \kappa_w\gamma^{-1}(\kappa^2 + m_w\kappa^5\kappa_b^2(1-\gamma)^{-1}) + \kappa_r\sum_{z=0}^{l-1}\kappa_z + \kappa_r\gamma^{-1}(1-\gamma)^{l-1}m_r\kappa^5\kappa_b^2
\end{aligned}
$$

where we have used the fact that $\sum_{n=0}^{N}(1-\gamma)^n \leq \frac{1}{\gamma}$ in the last inequality.

**Lemma 7** *Define* $Y_{H,k} = [Y_1, ..., Y_t, ..., Y_{2H}] = [M_{k-H:k} \ P_{k-H:k}]$ *and* $\tilde{Y}_{H,k} = [Y_1, ..., \tilde{Y}_t, ..., Y_{2H}]$ *where* $\tilde{Y}_{H,k}$ *has all its elements the same as* $Y_{H,k}$, *except one element. Then, the truncated cost function in equation 19 satisfies the following Lipschitz condition*

$$
|f_k(Y_1,,...,Y_t,...,Y_{2H}) - f_k(Y_1,,...,\tilde{Y}_t,...,Y_{2H})| \leq L_f\|Y_t - \tilde{Y}_t\|
$$

*where*

$$
L_f := 3G_cD\,\kappa_b\kappa^3(\kappa_r + \kappa_w). \tag{43}
$$

*Proof:* Based on Assumption 4, one has

$$
\begin{aligned}
&|f_k(Y_1,,...,Y_t,...,Y_{2H}) - f_k(Y_1,,...,\tilde{Y}_t,...,Y_{2H})| \\
&\leq G_cD\|\tilde{e}_k^\pi(Y_{H,k}) - \tilde{e}_k^\pi(\tilde{Y}_{H,k})\| + G_cD\|\tilde{u}_k^\pi(Y_{H,k}) - \tilde{u}_k^\pi(\tilde{Y}_{H,k})\|.
\end{aligned} \tag{44}
$$

where $\tilde{e}_k^\pi = \tilde{x}_k^\pi - r_k$. Using

$$
\tilde{e}_k^\pi(Y_{H,k}) - \tilde{e}_k^\pi(\tilde{Y}_{H,k}) = \tilde{x}_k^\pi(Y_{H,k}) - r_k + \tilde{x}_k^\pi(\tilde{Y}_{H,k}) - r_k = \tilde{x}_k^\pi(Y_{H,k}) - \tilde{x}_k^\pi(\tilde{Y}_{H,k})
$$

in equation 44 yields

$$
\begin{aligned}
&|f_k(Y_1,,...,Y_t,...,Y_{2H}) - f_k(Y_1,,...,\tilde{Y}_t,...,Y_{2H})| \\
&\leq G_cD\|\tilde{x}_k^\pi(Y_{H,k}) - \tilde{x}_k^\pi(\tilde{Y}_{H,k})\| + G_cD\|\tilde{u}_k^\pi(Y_{H,k}) - \tilde{u}_k^\pi(\tilde{Y}_{H,k})\|.
\end{aligned}
$$

Based on equation 17, if $Y_{H,k}$ and $\tilde{Y}_{H,k}$ differs in an $M_t$ element, one has

$$
\begin{aligned}
\|\tilde{x}_k^\pi(Y_{H,k}) - \tilde{x}_k^\pi(\tilde{Y}_{H,k})\| &= \|\tilde{A}_K^t B \sum_{i=0}^{m_w+H-1}\left(M_t^{i-t} - \tilde{M}_t^{i-t}\right)w_{k-i}\mathbb{I}_{1\leq i-t\leq m_w}\| \\
&\leq \kappa_b\kappa^2(1-\gamma)^t\kappa_w\sum_{i=1}^{2H}\|Y_t^{[i]} - \tilde{Y}_t^{[i]}\|.
\end{aligned} \tag{45}
$$

On the other hand, if $Y_{H,k}$ and $\tilde{Y}_{H,k}$ differs in an $P$ element, one has

$$
\begin{aligned}
\|\tilde{x}_k^\pi(Y_{H,k}) - \tilde{x}_k^\pi(\tilde{Y}_{H,k})\| &= \|\tilde{A}_K^t B \sum_{i=0}^{m_r+H-1}\left(P_t^{i-t} - \tilde{P}_t^{i-t}\right)r_{k-i}\mathbb{I}_{1\leq i-t\leq m_w}\| \\
&\leq \kappa_b\kappa^2(1-\gamma)^t\kappa_r\sum_{i=1}^{2H}\|Y_t^{[i]} - \tilde{Y}_t^{[i]}\|.
\end{aligned} \tag{46}
$$

Combining equation 45-equation 46, one has

$$\|\tilde{x}_k^\pi(Y_{H,k}) - \tilde{x}_k^\pi(\tilde{Y}_{H,k})\| \leq \kappa_b\kappa^2(1-\gamma)^t(\kappa_r + \kappa_w)\sum_{i=1}^{2H}\left(Y_t^{[i]} - \tilde{Y}_t^{[i]}\right) \leq \kappa_b\kappa^2(\kappa_r + \kappa_w)\|Y_t - \tilde{Y}_t\|. \tag{47}$$

On the other hand, based on equation 18, one has

$$\|\tilde{u}_k^\pi(Y_{H,k}) - \tilde{u}_k^\pi(\tilde{Y}_{H,k})\| = \|K(\tilde{x}_k^\pi(Y_{H,k}) - \tilde{x}_k^\pi(\tilde{Y}_{H,k})) + \sum_{i=1}^{m_w+m_r}\left(Y_t^{[i]} - \tilde{Y}_t^{[i]}\right)\|$$

$$\leq \left(\kappa_b\kappa^3(1-\gamma)^t(\kappa_r + \kappa_w) + 1\right)\sum_{i=1}^{2H}\|Y_t^{[i]} - \tilde{Y}_t^{[i]}\| \leq 2\kappa_b\kappa^3(\kappa_r + \kappa_w))\|Y_t - \tilde{Y}_t\|$$

$$\tag{48}$$

where the first equality is obtained based on the fact that $K$ is $(\kappa, \gamma)$-stable (see Definition 1). Using equation 47 and equation 48 in equation 44 completes the proof.

**Lemma 8** *Let Assumption 5 is satisfied. Then, the following gradient bound is satisfied*

$$\|\nabla_{Y_{H,k}}f_k(Y_{H,k})\|_F \leq 6Hd^2G_c\left(\kappa_r + \kappa_w\right)\kappa_b\kappa^3\gamma^{-1}G_f \tag{49}$$

*where $d = \max(n, m)$.*

*Proof:* To prove the claim, We bound $\nabla_{Y_{p,q}^{[l]}}f_k(Y_{H,k})$ for every $p \in \{1, ..., m\}$, $q \in \{1, ..., n\}$ and $l \in \{1, ..., 2H\}$. We find the bound for the two cases: when $Y_{p,q}^{[l]} = M_{p,q}^{[l_1]}$, for which $l_1 \in \{1, ..., m_w\}$, and when $Y_{p,q}^{[l_2]} = P_{p,q}^{[l]}$, for which $l_2 \in \{1, ..., m_r\}$. For the first case, similar to Zhao et al. (2022), one can show that

$$|\nabla_{M_{p,q}^{[l]}}f_k(Y_{H,k})| \leq 3G_c\kappa_w\kappa_b\kappa^3\gamma^{-1}. \tag{50}$$

For the second case, using the same procedure as in Zhao et al. (2022), one can show that

$$|\nabla_{P_{p,q}^{[l]}}f_k(Y_{H,k})| \leq 3G_c\kappa_r\kappa_b\kappa^3\gamma^{-1}. \tag{51}$$

Therefore, since $pq \leq \max(n, m)^2 = d^2$, and $l_1 + l_2 = 2H$, one has $\|\nabla_{Y_{H,k}}f_k(Y_{H,k})\|_F \leq 6Hd^2G_c(\kappa_r + \kappa_w)\kappa_b\kappa^3\gamma^{-1}$.

## E   Proof of Theorem 2

The solution to the system in equation 1 using the linear feedback controller in equation 3 is given in equation 32. Using equation 32, the linear feedback controller in equation 3 reads

$$u_k^{\text{lin}} = K_{fb}x_k^{\text{lin}} + K_{ff}\sum_{j=0}^{l-1}N^{[l-j]}r_{k-j}$$

$$= \sum_{j=1}^{k}\sum_{q=0}^{\min(l-1,j-1)}K_{fb}(A + BK_{fb})^{j-q-1}BK_{ff}N^{[l-q]}r_{k-j}$$

$$+ \sum_{i=1}^{k}K_{fb}(A + BK_{fb})^{i-1}w_{k-i} + \sum_{j=0}^{l-1}K_{ff}N^{[l-j]}r_{k-j}. \tag{52}$$

We aim to approximate equation 52 with a memory-augmented policy in equation 12. Replacing equation 32 in the memory-augmented policy equation 12, we get

$$
\begin{aligned}
u_k^\pi =& Kx_k + \sum_{i=1}^{m_w} M^{[i-1]} w_{k-i} + \sum_{j=0}^{m_r-1} P^{[j]} r_{k-j} \\
=& \sum_{j=1}^{k} \sum_{q=0}^{\min(l-1,j-1)} K(A+BK_{fb})^{j-q-1} BK_{ff} N^{[l-q]} r_{k-j} \\
&+ \sum_{i=1}^{k} K(A+BK_{fb})^{i-1} w_{k-i} + \sum_{i=1}^{m_w} M^{[i-1]} w_{k-i} + \sum_{j=0}^{m_r-1} P^{[j]} r_{k-j}.
\end{aligned}
\tag{53}
$$

Now, we derive $u_k^{\text{lin}} - u_k^\pi$

$$
\begin{aligned}
u_k^{\text{lin}} - u_k^\pi =& \sum_{i=1}^{m_w} [(K_{fb}-K)(A+BK_{fb})^{i-1} - M^{[i-1]}] w_{k-i} \\
&+ \sum_{i=m_w+1}^{k} (K_{fb}-K)(A+BK_{fb})^{i-1} w_{k-i} \\
&+ \sum_{j=1}^{k} \sum_{q=0}^{\min(l-1,j-1)} (K_{fb}-K)(A+BK_{fb})^{j-q-1} BK_{ff} N^{[l-q]} r_{k-j} \\
&+ \sum_{j=0}^{l-1} K_{ff} N^{[l-j]} r_{k-j} - \sum_{j=0}^{m_r-1} P^{[j]} r_{k-j}.
\end{aligned}
\tag{54}
$$

Select $M_w^{[i-1]} = (K_{fb}-K)(A+BK_{fb})^{i-1}$ for $i=1,...,m_w$. This makes the coefficients of $w_{k-i}$ equal to zero for $i=1,...,m_w$.

Similarly, we try to make the coefficients of $r_{k-j}$ equal to zero. We have three cases. a) $j=0$. $r_k$ appears in the fourth line of equation 54 and its coefficient becomes zero by selecting $P^{[0]} = K_{ff} N^{[l]}$. b) $0 < j < l$. In this case, $r_{k-j}$ appears in the third and fourth lines of equation 54 and $\min(l-1, j-1) = j-1$. Setting the coefficient of $r_{k-j}$ equal to zero

$$
\sum_{q=0}^{j-1} (K_{fb}-K)(A+BK_{fb})^{j-q-1} BK_{ff} N^{[l-q]} + K_{ff} N^{[l-j]} - P^{[j]} = 0,
$$

we have

$$
P^{[j]} = \sum_{q=0}^{j-1} (K_{fb}-K)(A+BK_{fb})^{j-q-1} BK_{ff} N^{[l-q]} + K_{ff} N^{[l-j]}.
$$

Finally c) $l \leq j < m_r$. In this case, $r_{k-j}$ appears in the third and forth lines of equation 54 and $\min(l-1, j-1) = l-1$. Setting the coefficient of $r_{k-j}$ equal to zero

$$
\sum_{q=0}^{l-1} (K_{fb}-K)(A+BK_{fb})^{j-q-1} BK_{ff} N^{[l-q]} - P^{[j]} = 0,
$$

we have

$$
P^{[j]} = \sum_{q=0}^{l-1} (K_{fb}-K)(A+BK_{fb})^{j-q-1} BK_{ff} N^{[l-q]}.
$$

The aforementioned results are summarized in equation 20. If we select $M$, $P$ according to equation 20, then $u_k^{\text{lin}} - u_k^\pi$ reads,

$$
\begin{aligned}
u_k^{\text{lin}} - u_k^\pi &= \sum_{i=m_w+1}^{k} (K_{fb} - K)(A + BK_{fb})^{i-1} w_{k-i} + \sum_{j=m_r}^{k} \sum_{q=0}^{l-1} (K_{fb} - K)(A + BK_{fb})^{j-q-1} BK_{ff} N^{[l-q]} r_{k-j} \\
&= \sum_{i=m_w+1}^{k} M^{[i-1]} w_{k-i} + \sum_{j=m_r}^{k} P^{[j]} r_{k-j}.
\end{aligned}
\tag{55}
$$

As a result

$$
\begin{aligned}
\|u_k^{\text{lin}} - u_k^\pi\| &\leq \sum_{i=m_w+1}^{k} \kappa_b \kappa^3 (1-\gamma)^{i-1} \kappa_w + \sum_{j=m_r}^{k} \kappa_b \kappa^3 (1-\gamma)^j \kappa_r \\
&= \sum_{t=0}^{k-m_w-1} \kappa_b \kappa^3 (1-\gamma)^{t+m_w} \kappa_w + \sum_{s=0}^{k-m_r} \kappa_b \kappa^3 (1-\gamma)^{s+m_r} \kappa_r \\
&\leq \gamma^{-1} \kappa_b \kappa^3 (1-\gamma)^{m_w} \kappa_w + \gamma^{-1} \kappa_b \kappa^3 (1-\gamma)^{m_r} \kappa_r.
\end{aligned}
$$

where we changed the indices in the summations as $t = i - m_w - 1$ and $s = j - m_r$ to get the equality and used the fact that $\sum_{n=0}^{N} (1-\gamma)^n \leq \frac{1}{\gamma}$ to get the last inequality.

## F   Proof of Theorem 3

Using $u_k^{\text{lin}}$ in equation 3, $x_{k+1}^{\text{lin}}$ can be written as

$$
x_{k+1}^{\text{lin}} = A x_k^{\text{lin}} + B u_k^{\text{lin}} + w_k = \tilde{A}_K x_k^{\text{lin}} + B(K_{fb} - K) x_k^{\text{lin}} + BK_{ff} \sum_{j=0}^{l-1} N^{[l-j]} r_{k-j} + w_k.
\tag{56}
$$

Using $u_k^\pi$ in equation 12, $x_{k+1}^\pi$ can be written as

$$
x_{k+1}^\pi = \tilde{A}_K x_k^\pi + B \sum_{t=1}^{m_w} M^{[t-1]} w_{k-t} + w_k + B \sum_{s=0}^{m_r-1} P^{[s]} r_{k-s}.
\tag{57}
$$

Based on equation 56-equation 57

$$
\begin{aligned}
x_{k+1}^{\text{lin}} - x_{k+1}^\pi &= \tilde{A}_K (x_k^{\text{lin}} - x_k^\pi) + B(K_{fb} - K) x_k^{\text{lin}} + BK_{ff} \sum_{j=0}^{l-1} N^{[l-j]} r_{k-j} \\
&\quad - B \sum_{t=1}^{m_w} M^{[t-1]} w_{k-t} - B \sum_{s=0}^{m_r-1} P^{[s]} r_{k-s}.
\end{aligned}
$$

Replace $x_k^{\text{lin}}$ from equation 32 in $B(K_{fb} - K) x_k^{\text{lin}}$

$$
\begin{aligned}
x_{k+1}^{\text{lin}} - x_{k+1}^\pi &= \tilde{A}_K (x_k^{\text{lin}} - x_k^\pi) + B(K_{fb} - K) \sum_{j=1}^{k} \sum_{q=0}^{\min(l-1, j-1)} (A + BK_{fb})^{j-q-1} BK_{ff} N^{[l-q]} r_{k-j} \\
&\quad + B(K_{fb} - K) \sum_{i=1}^{k} (A + BK_{fb})^{i-1} w_{k-i} + BK_{ff} \sum_{j=0}^{l-1} N^{[l-j]} r_{k-j} \\
&\quad - B \sum_{t=1}^{m_w} M^{[t-1]} w_{k-t} - B \sum_{s=0}^{m_r-1} P^{[s]} r_{k-s}.
\end{aligned}
\tag{58}
$$

Using $M_w$ in equation 20, the terms containing $w$ in the above equation read

$$t_w = B(K_{fb} - K) \sum_{i=1}^{k} (A + BK_{fb})^{i-1} w_{k-i} - B \sum_{t=1}^{m_w} M^{[t-1]} w_{k-t}$$

$$= B(K_{fb} - K) \sum_{i=1}^{k} (A + BK_{fb})^{i-1} w_{k-i} - B(K_{fb} - K) \sum_{t=1}^{m_w} (A + BK_{fb})^{t-1} w_{k-t}$$

$$= B(K_{fb} - K) \sum_{t=m_w+1}^{k} (A + BK_{fb})^{t-1} w_{k-t}.$$

Using $P$ in equation 20, the terms containing $r$ in the above equation read

$$t_r = B(K_{fb} - K) \sum_{j=1}^{k} \sum_{q=0}^{\min(l-1,j-1)} (A + BK_{fb})^{j-q-1} BK_{ff} N^{[l-q]} r_{k-j} + BK_{ff} \sum_{j=0}^{l-1} N^{[l-j]} r_{k-j}$$

$$- B \sum_{s=0}^{m_r-1} P^{[s]} r_{k-s} = B(K_{fb} - K) \sum_{j=1}^{k} \sum_{q=0}^{\min(l-1,j-1)} (A + BK_{fb})^{j-q-1} BK_{ff} N^{[l-q]} r_{k-j}$$

$$+ BK_{ff} \sum_{j=0}^{l-1} N^{[l-j]} r_{k-j} - B(K_{fb} - K) \sum_{s=1}^{m_r-1} \sum_{q=0}^{\min(l-1,s-1)} (A + BK_{fb})^{s-q-1} BK_{ff} N^{[l-q]} r_{k-s}$$

$$- BK_{ff} \sum_{s=0}^{l-1} N^{[l-s]} r_{k-s} = B(K_{fb} - K) \sum_{s=m_r}^{k} \sum_{q=0}^{l-1} (A + BK_{fb})^{s-q-1} BK_{ff} N^{[l-q]} r_{k-s}.$$

Using $t_w$, $t_r$ in equation 58, $x_{k+1}^{\text{lin}} - x_{k+1}^{\pi}$ reads

$$x_{k+1}^{\text{lin}} - x_{k+1}^{\pi} = \tilde{A}_K (x_k^{\text{lin}} - x_k^{\pi}) + B(K_{fb} - K) \sum_{t=m_w+1}^{k} (A + BK_{fb})^{t-1} w_{k-t}$$

$$+ B(K_{fb} - K) \sum_{s=m_r}^{k} \sum_{q=0}^{l-1} (A + BK_{fb})^{s-q-1} BK_{ff} N^{[l-q]} r_{k-s}.$$

Since $\tilde{A}_K$ is stable, $x_k^{\text{lin}} - x_k^{\pi}$ reads

$$x_k^{\text{lin}} - x_k^{\pi} = \sum_{i=1}^{k} \tilde{A}_K^{i-1} B(K_{fb} - K) \sum_{t=m_w+1}^{k-i} (A + BK_{fb})^{t-1} w_{k-i-t}$$

$$+ \sum_{i=1}^{k} \tilde{A}_K^{i-1} B(K_{fb} - K) \sum_{s=m_r}^{k-i} \sum_{q=0}^{l-1} (A + BK_{fb})^{s-q-1} BK_{ff} N^{[l-q]} r_{k-i-s}.$$

Since it should hold that $k - i \geq m_w + 1$ in the second line and $k - i \geq m_r$ in the third line,

$$x_k^{\text{lin}} - x_k^{\pi} = \sum_{i=1}^{k-m_w-1} \tilde{A}_K^{i-1} B(K_{fb} - K) \sum_{t=m_w+1}^{k-i} (A + BK_{fb})^{t-1} w_{k-i-t}$$

$$+ \sum_{i=1}^{k-m_r} \tilde{A}_K^{i-1} B(K_{fb} - K) \sum_{s=m_r}^{k-i} \sum_{q=0}^{l-1} (A + BK_{fb})^{s-q-1} BK_{ff} N^{[l-q]} r_{k-i-s}.$$

Using equation 20, we have

$$x_k^{\text{lin}} - x_k^{\pi} = \sum_{i=1}^{k-m_w-1} \tilde{A}_K^{i-1} B \sum_{t=m_w+1}^{k-i} M^{[t-1]} w_{k-i-t} + \sum_{i=1}^{k-m_r} \tilde{A}_K^{i-1} B \sum_{s=m_r}^{k-i} P^{[s]} r_{k-i-s}. \tag{59}$$

Based on $(\kappa, \gamma)$-stability of $K$ and Assumption 5

$$
\begin{aligned}
\|x_k^{\lin} - x_k^\pi\| &\leq \sum_{i=1}^{k-m_w-1} \kappa^2 \kappa_b (1-\gamma)^{i-1} \sum_{t=m_w+1}^{k-i} \kappa_b \kappa^3 (1-\gamma)^{t-1} \kappa_w \\
&\quad + \sum_{i=1}^{k-m_r} \kappa^2 \kappa_b (1-\gamma)^{i-1} \sum_{s=m_r}^{k-i} \kappa_b \kappa^3 (1-\gamma)^s \kappa_r \\
&= \sum_{i=1}^{k-m_w-1} \kappa^2 \kappa_b (1-\gamma)^{i-1} \sum_{p=0}^{k-i-m_w-1} \kappa_b \kappa^3 (1-\gamma)^{p+m_w} \kappa_w \\
&\quad + \sum_{i=1}^{k-m_r} \kappa^2 \kappa_b (1-\gamma)^{i-1} \sum_{q=0}^{k-i-m_r} \kappa_b \kappa^3 (1-\gamma)^{q+m_r} \kappa_r \\
&\leq \sum_{i=1}^{k-m_w-1} \kappa^2 \kappa_b (1-\gamma)^{i-1} \gamma^{-1} \kappa_b \kappa^3 (1-\gamma)^{m_w} \kappa_w \\
&\quad + \sum_{i=1}^{k-m_r} \kappa^2 \kappa_b (1-\gamma)^{i-1} \gamma^{-1} \kappa_b \kappa^3 (1-\gamma)^{m_r} \kappa_r \\
&\leq \gamma^{-2} \kappa_b^2 \kappa^5 (1-\gamma)^{m_w} \kappa_w + \gamma^{-2} \kappa_b^2 \kappa^5 (1-\gamma)^{m_r} \kappa_r.
\end{aligned}
$$

where we changed the indices in the summations as $p = t - m_w - 1$ and $q = s - m_r$ to get the equality and used the fact that $\sum_{n=0}^{N} (1-\gamma)^n \leq \frac{1}{\gamma}$ in the last two inequalities.

## G  Proof of Theorem 4

Before proving the regret bound, we present Lemma 9 which provides an upper bound for the difference between the costs using optimal linear controller and optimal memory-augmented control policy.

**Lemma 9** *Let Assumptions 1-5 hold. Let $K_f^* = [K_{fb}^* \ K_{ff}^*]$ denote the optimal linear gain. Let $x_k^{lin}(K_f^*)$ denote the state using the optimal linear controller $u_k^{lin}(K_f^*)$. Set $H = m_w = m_r$ and let $Y^* = [M^{[0]*}, ..., M^{[H-1]*}, P^{[0]*}, ..., P^{[H-1]*}]$ denote the optimal weights learned by Algorithm 1. Let $\tilde{x}_k^\pi(Y^*), \tilde{u}_k^\pi(Y^*)$ denote the truncated state and control using the optimal weights according to equation 17-equation 18. Then*

$$
|c_k(\tilde{x}_k^\pi(Y^*) - r_k, \tilde{u}_k^\pi(Y^*)) - c_k(x_k^{lin}(K_f^*) - r_k, u_k^{lin}(K_f^*))| \leq 2 G_c D \gamma^{-1} H (\kappa_w + \kappa_r) \kappa^6 \kappa_b^2 (1-\gamma)^{(H-1)}. \tag{60}
$$

*Proof of Lemma 9:*

By selecting $K = K_{fb}^*$, $M = \mathbf{0}$ and $P^{[s]} = K_{ff}^* N^{[l-s]}$, $0 \leq s < l$, equation 16 can be used to express $x_k^{lin}(K_f^*)$

$$
\begin{aligned}
x_k^{lin}(K_f^*) &= \sum_{y=0}^{k-1} \Psi_{k,y}^{K_{fb}^*,k} w_{k-y-1} + \sum_{z=0}^{k-1} \psi_{k,z}^{K_{fb}^*,k} r_{k-z} = \sum_{y=0}^{H-1} \Psi_{k,y}^{K_{fb}^*,k} w_{k-y-1} + \sum_{z=0}^{H-1} \psi_{k,z}^{K_{fb}^*,k} r_{k-z} \\
&\quad + \sum_{y=H}^{k-1} \Psi_{k,y}^{K_{fb}^*,k} w_{k-y-1} + \sum_{z=H}^{k-1} \psi_{k,z}^{K_{fb}^*,k} r_{k-z} = \sum_{y=H}^{k-1} \Psi_{k,y}^{K_{fb}^*,k} w_{k-y-1} + \sum_{z=H}^{k-1} \psi_{k,z}^{K_{fb}^*,k} r_{k-z} + x_H^{lin}(K_f^*).
\end{aligned}
$$

As a result, $\|x_k^{lin}(K_f^*) - \tilde{x}_k^\pi(Y^*)\|$ reads

$$
\|x_k^{lin}(K_f^*) - \tilde{x}_k^\pi(Y^*)\| \leq \|x_H^{lin}(K_f^*) - \tilde{x}_H^\pi(Y^*)\| + \|\sum_{y=H}^{k-1} \Psi_{k,y}^{K_{fb}^*,k} w_{k-y-1}\| + \|\sum_{z=H}^{k-1} \psi_{k,z}^{K_{fb}^*,k} r_{k-z}\|.
$$

By equation 59, $\|x_H^{lin}(K_f^*) - \tilde{x}_H^\pi(Y^*)\| = 0$. By Lemma 5

$$\|\sum_{y=H}^{k-1} \Psi_{k,y}^{K_{fb}^*,k} w_{k-y-1}\| \le \sum_{y=H}^{k-1} H\kappa^5 \kappa_b^2 (1-\gamma)^{y-1} \kappa_w \le \gamma^{-1} H\kappa^5 \kappa_b^2 (1-\gamma)^{H-1} \kappa_w,$$

$$\|\sum_{y=H}^{k-1} \psi_{k,z}^{K_{fb}^*,k} r_{k-z}\| \le \sum_{y=H}^{k-1} H\kappa^5 \kappa_b^2 (1-\gamma)^{y-1} \kappa_r \le \gamma^{-1} H\kappa^5 \kappa_b^2 (1-\gamma)^{H-1} \kappa_r. \tag{61}$$

Hence,

$$\|x_k^{lin}(K_f^*) - \tilde{x}_k^\pi(Y^*)\| \le \gamma^{-1} H(\kappa_w + \kappa_r)\kappa^5 \kappa_b^2 (1-\gamma)^{H-1}.$$

Next, we find $\|u_k^{lin}(K_f^*) - \tilde{u}_k^\pi(Y^*)\|$. In equation 18, set $K \equiv K_{fb}^*$. Then, one has

$$u_k^{lin}(K_f^*) = K_{fb}^* x_k^{lin} + \sum_{s=0}^{H-1} P^* r_{k-s} = \sum_{y=H}^{k-1} K_{fb}^* \Psi_{k,y}^{K_{fb}^*,k} w_{k-y-1} + \sum_{z=H}^{k-1} K_{fb}^* \psi_{k,z}^{K_{fb}^*,k} r_{k-z} + K_{fb}^* x_H^{lin} + \sum_{s=0}^{H-1} P^* r_{k-s}$$

$$= \sum_{y=H}^{k-1} K_{fb}^* \Psi_{k,y}^{K_{fb}^*,k} w_{k-y-1} + \sum_{z=H}^{k-1} K_{fb}^* \psi_{k,z}^{K_{fb}^*,k} r_{k-z} + u_H^{lin}(K_f^*).$$

As a result, one can write

$$\|u_k^{lin}(K_f^*) - \tilde{u}_k^\pi(Y^*)\| \le \|u_H^{lin}(K_f^*) - \tilde{u}_k^\pi(Y^*)\| + \sum_{y=H}^{k-1} \|K_{fb}^* \Psi_{k,y}^{K_{fb}^*,k} w_{k-y-1}\| + \sum_{z=H}^{k-1} \|K_{fb}^* \psi_{k,z}^{K_{fb}^*,k} r_{k-z}\|.$$

By equation 55, $\|u_H^{lin}(K_f^*) - \tilde{u}_k^\pi(Y^*)\| = 0$. Using equation 61

$$\|u_k^{lin}(K_f^*) - \tilde{u}_k^\pi(Y^*)\| \le \gamma^{-1} H(\kappa_w + \kappa_r)\kappa^6 \kappa_b^2 (1-\gamma)^{H-1}.$$

Therefore,

$$|c_k(\tilde{x}_k^\pi(Y^*) - r_k, \tilde{u}_k^\pi(Y^*)) - c_k(x_k^{lin}(K_f^*) - r_k, u_k^{lin}(K_f^*))|$$

$$\le G_c D \|x_k^{lin}(K_f^*) - \tilde{x}_k^\pi(Y^*)\| + G_c D \|u_k^{lin}(K_f^*) - \tilde{u}_k^\pi(Y^*)\|$$

$$\le 2 G_c D \gamma^{-1} H(\kappa_w + \kappa_r)\kappa^6 \kappa_b^2 (1-\gamma)^{(H-1)}.$$

*Proof of Theorem 4:* The regret reads

$$\text{Regret} = \sum_{k=1}^T c_k(e_k, u_k) - \min_{K_f \in \mathcal{K}} \sum_{k=1}^T c_k(e_k, u_k) \tag{62}$$

$$= \underbrace{\sum_{k=1}^T c_k(e_k(Y_{0:k-1}), u_k(Y_{0:k-1})) - \sum_{k=1}^T f_k(Y_{H,k})}_{\alpha_T}$$

$$+ \underbrace{\sum_{k=1}^T f_k(Y_{H,k}) - \sum_{k=1}^T f_k(Y^*)}_{\beta_T}$$

$$+ \underbrace{\sum_{k=1}^T f_k(Y^*) - \min_{K_f \in \mathcal{K}} \sum_{k=1}^T c_k(e_k, u_k)}_{\zeta_T}$$

where $Y^* = [M^{[0]*}, ..., M^{[H-1]*}, P^{[0]*}, ..., P^{[H-1]*}] \in (\mathbb{R}^{m \times n})^{2H}$ denote the optimal weights learned by Algorithm 1 satisfying Assumption 5.

The regret analysis is split into three parts: $\alpha_T$ denotes the difference between the cost of Algorithm 1 and the truncated cost. $\beta_T$ denotes the difference between the truncated and optimal truncated costs. $\zeta_T$ denotes the difference between the optimal truncated cost and the optimal linear control policy.

We now bound the first term $\alpha_T$. One has

$$|c_k(e_k, u_k) - f_k(Y_{H,k})| \leq G_c\,D\,\|(x_k^K(Y_{0:k-1}) - r_k) - (\tilde{x}_k^\pi(Y_{H,k}) - r_k)\| + G_c D \|u_k^K(Y_{0:k-1}) - \tilde{u}_k^\pi(Y_{H,k})\|$$
$$\leq 2G_c D^2 \kappa^3 (1-\gamma)^H$$

where we have used Lemma 5 to get the above result. Therefore,

$$\|\alpha_T\| = \|\sum_{k=1}^{T} c_k(e_k, u_k) - \sum_{k=1}^{T} f_k(Y_{H,k})\| \leq 2T\,G_c D^2 \kappa^3 (1-\gamma)^H = \mathcal{O}(\sqrt{T}) \tag{63}$$

where the last equality is obtained based on $H = \mathcal{O}(\log T)$.

We can bound the term $\beta_T$ by Theorem 4.6 of Agarwal et al. (2019) and the results of Lemmas 7 and 8 as

$$\sum_{k=1}^{T} f_k(Y_{H,k}) - \sum_{k=1}^{T} f_k(Y^*) \leq \frac{1}{\eta} M_b^2 + T G_f^2 \eta + L_f H^2 \eta G_f T \tag{64}$$

where $M_b := 2\sqrt{d}\kappa_b \kappa^3 \gamma^{-1}$, $d = \max(n, m)$. By selecting $\eta = \mathcal{O}(\frac{1}{\sqrt{T}})$, $H = \mathcal{O}(\log T)$, we have $\beta_T = \mathcal{O}(\sqrt{T})$.

We now bound the third term $\zeta_T$. Based on equation 63, one has,

$$\|\sum_{k=1}^{T} f_k(Y^*) - \sum_{k=1}^{T} c_k(x_k^{lin}(K_f*) - r_k, u_k^{lin}(K_f*))\|$$
$$\leq \|\sum_{k=1}^{T} c_k(\tilde{x}_k^\pi(Y^*) - r_k, \tilde{u}_k^\pi(Y^*)) - \sum_{k=1}^{T} c_k(x_k^{lin}(K_f^*) - r_k, u_k^{lin}(K_f^*))\| + 2T\,G_c D^2 \kappa^3 (1-\gamma)^H.$$

Using Lemma 9,

$$\|\sum_{k=1}^{T} f_k(Y^*) - \sum_{k=1}^{T} c_k(x_k^{lin}(K_f*) - r_k, u_k^{lin}(K_f*))\| \leq 2TG_c D\gamma^{-1} H(\kappa_w + \kappa_r)\kappa^6 \kappa_b^2 (1-\gamma)^{(H-1)}$$
$$+ 2T\,G_c D^2 \kappa^3 (1-\gamma)^H = \mathcal{O}(\sqrt{T}) \tag{65}$$

where the last equality is obtained based on $H = \mathcal{O}(\log T)$.

## H  Additional simulation results for changing reference generator dynamics

Our proof of the regret bound in Theorem 4, is independent of the dynamics of the reference signal generator $(S, F)$. We make Assumption 3, stating that the reference signal is bounded and the output of the reference signal is measurable. We also assume that the classical state-tracking problem is solvable, such that the problem setup is meaningful. Since the information of $(S, F)$ is not used, we expect that our algorithm works even if $(S, F)$ changes at some steps. Theoretical guarantees for changing $(S, F)$ however require careful examination of the theories to define the minimum time intervals between dynamics changes and it is out of the scope of this paper. In the sequel, we give some simulation results to show the performance of Algorithm 1 against the other control approaches in Section 6.3.

We consider the same problem setup as Section 6, including the same types of disturbances. We assume that the dynamics of the reference generator changes at $T_{\text{change}} = 8000$ from $(S_1, F_1)$ in equation 26 to $(S_2, F_2)$

$$
z_{k+1} = \begin{bmatrix} 0 & 1 & 0 \\ -1 & 0 & 0 \\ 0 & 0 & 0 \end{bmatrix} z_k, \ z_{T_{\text{change}}} = [1, 0, -1]^T,
$$
$$
r_k = \begin{bmatrix} 0 & 0 & -1 \\ 1 & 0 & 1 \end{bmatrix} z_k.
$$
(66)

All control approaches in Section 6.3, namely the Online control algorithm with a fixed feedforward gain, LQR, and $H_\infty$-control, require a method to detect when the dynamics of the reference generator changes to learn the new dynamics and compute the new feedforward gain. However, our algorithm does not need such a detection algorithm. In the simulation result, we have assumed the Online control algorithm with a fixed feedforward gain, LQR, and $H_\infty$-control are aware of $T_{\text{change}}$. Let $\Gamma_1$, $\Pi_1$ be the solution to the tracking equation in equation 5 using $(S_1, F_1)$ and $\Gamma_2$, $\Pi_2$ be the solution to the tracking equation in equation 5 using $(S_2, F_2)$. At the beginning of the simulation, the feedforward gain in the Online control algorithm with a fixed feedforward gain is set to $K_{ff1} = \Gamma_1 - KF_1$. Similarly, we set $K_{ff1} = \Gamma_1 - K_{fb}F_1$ in the LQR, and $H_\infty$-control approaches. At $k = T_{\text{change}}$, we change the feedforward gain according to the new dynamics to $K_{ff2} = \Gamma_2 - KF_2$ and $K_{ff2} = \Gamma_2 - K_{fb}F_2$ in the Online control algorithm with a fixed feedforward gain, and the LQR and $H_\infty$-control approaches. Algorithm 1 is run as discussed in Section 6.3; i.e. no information of $T_{\text{change}}$ or $(S_1, F_1)$, $(S_2, F_2)$ is used.

Table 2: The final average costs suffered by the algorithms for running each algorithm for $T = 16000$ steps. At $k = T_{\text{change}}$, the dynamics of the reference generator changes from $(S_1, F_1)$ to $(S_2, F_2)$. Bold values show the lowest average cost for each case of disturbance. The algorithm LQR for random walk is the optimal controller in the case of random walk disturbance and thus it is only evaluated in this case.

| **Disturbance** | Algorithm 1 | Online control with fixed feedforward gain | LQR | $H_\infty$ | LQR for random walk |
|---|---|---|---|---|---|
| Gaussian | 3.83 | 3.74 | **3.65** | 5.25 | N.A. |
| Random walk | 16.65 | 36.39 | 283.79 | 109.01 | **16.12** |
| Uniformly sam. | **8.09** | 10.76 | 19.88 | 23.25 | N.A. |
| Constant | **5.82** | 8.65 | 55.09 | 17.30 | N.A. |
| Amplitude mod. | **4.10** | 4.17 | 15.91 | 6.58 | N.A. |
| Sinusoidal | 4.52 | **4.51** | 28.53 | 10.00 | N.A. |

In Table 2, we summarize the final average cost; $\frac{1}{T} \sum_{k=1}^{T} c_k(e_k, u_k)$ suffered by the algorithms for $T = 16000$. We use **bold** to refer to the algorithm with the lowest average cost in each case of the disturbance.

In Fig. 1, the evolution of the average cost $J_T = \frac{1}{T} \sum_{k=1}^{T} c_k(e_k, u_k)$ in equation 10 vs. $T$ for the 6 cases of disturbance in subsection 6.2 has been shown. The discussion regarding the performance of the algorithms is similar to Section 6.2 and thus omitted.

We plot the reference trajectories and the tracking errors for $T_{\text{change}} - 30$ to $T_{\text{change}}$ and $T - 30$ to $T$ in Fig. 10-16. These time intervals are selected such that one can see the performance of Algorithm 1 and the online control with a fixed feedforward gain when those algorithms converge.

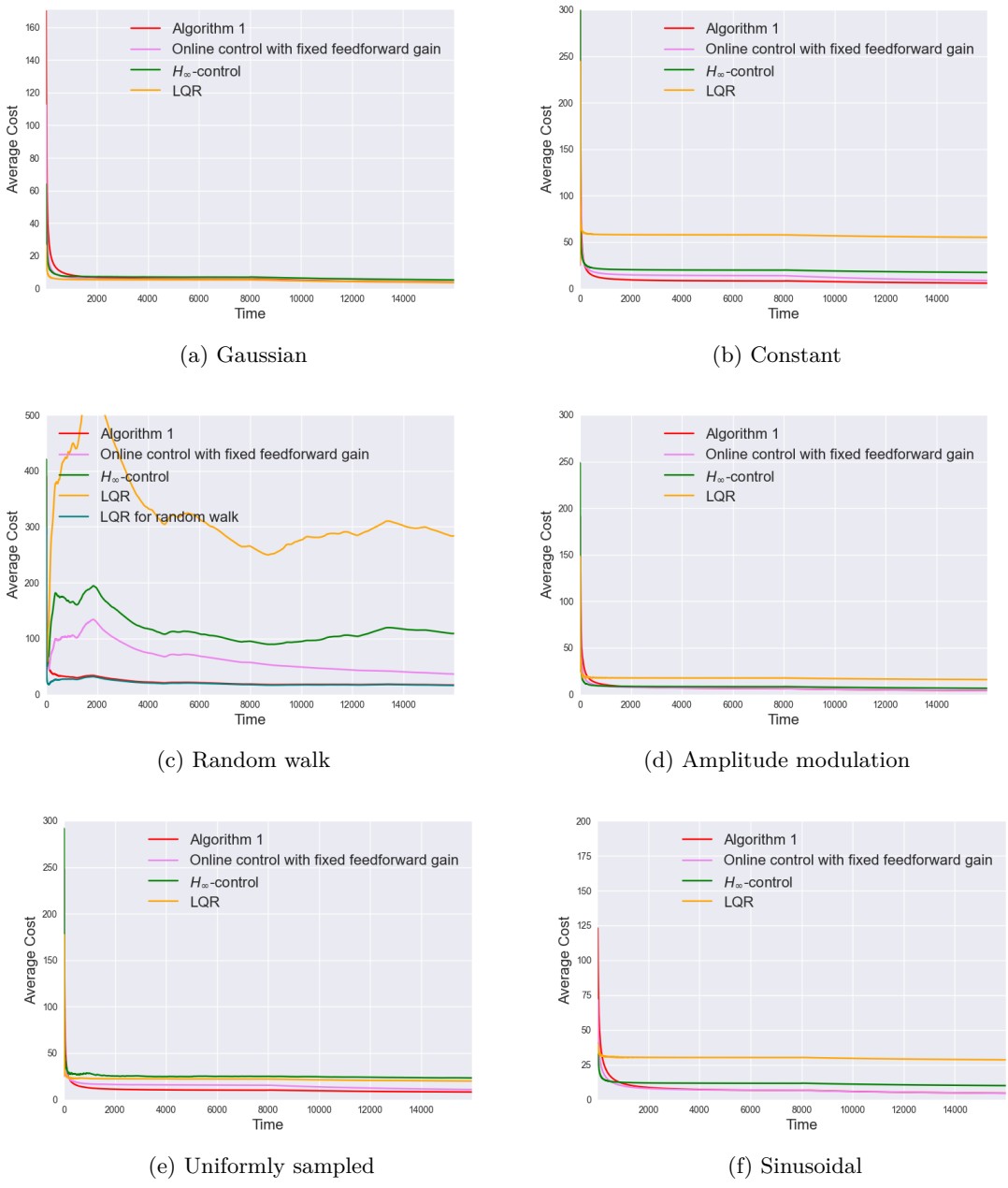

Figure 9: The evolution of the average cost $J_T = \frac{1}{T}\sum_{k=1}^{T} c_k(e_k, u_k)$ in equation 10 vs. $T$ for the presented Algorithm 1, versus, the online control algorithm with a fixed feedforward gain, the $H_\infty$ control and the LQR for Gaussian, random walk, uniformly sampled, constant, amplitude modultion and sinusoidal disturbances. At $k = T_{\text{change}}$, the dynamics of the reference generator changes from $(S_1, F_1)$ to $(S_2, F_2)$. In each case, we generate the disturbance in the beginning of the simulation so the disturbance sequence is the same for all algorithms. Details regarding disturbances are given in Subsection 6.2. The final values of the average costs are reported in Table 2.

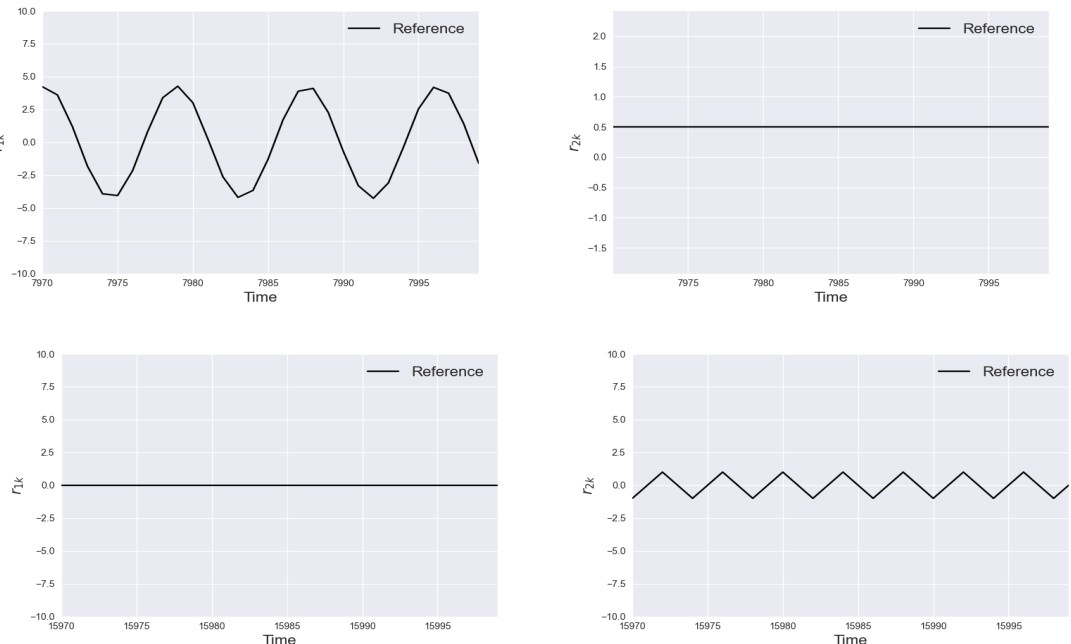

Figure 10: The reference signals over $T_{\text{change}} - 30$ to $T_{\text{change}}$ and $T - 30$ to $T$.

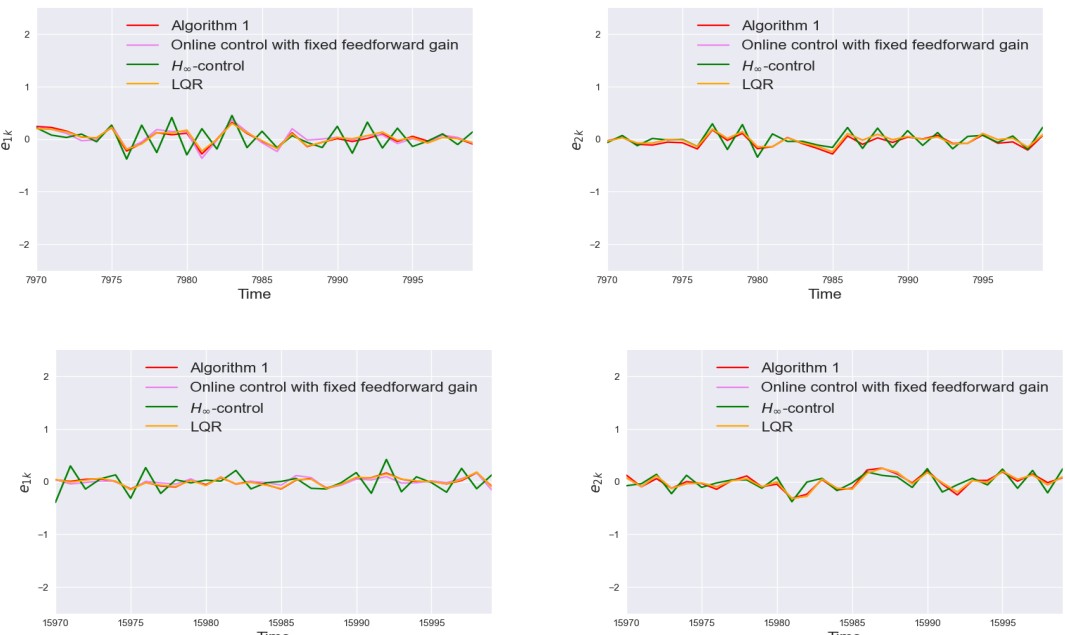

Figure 11: Tracking error over $T_{\text{change}} - 30$ to $T_{\text{change}}$ and $T - 30$ to $T$ for the Gaussian disturbance for the presented Algorithm 1, versus, the online control algorithm with a fixed feedforward gain, the $H_\infty$ control and the LQR control using the reference signals in Fig. 10.

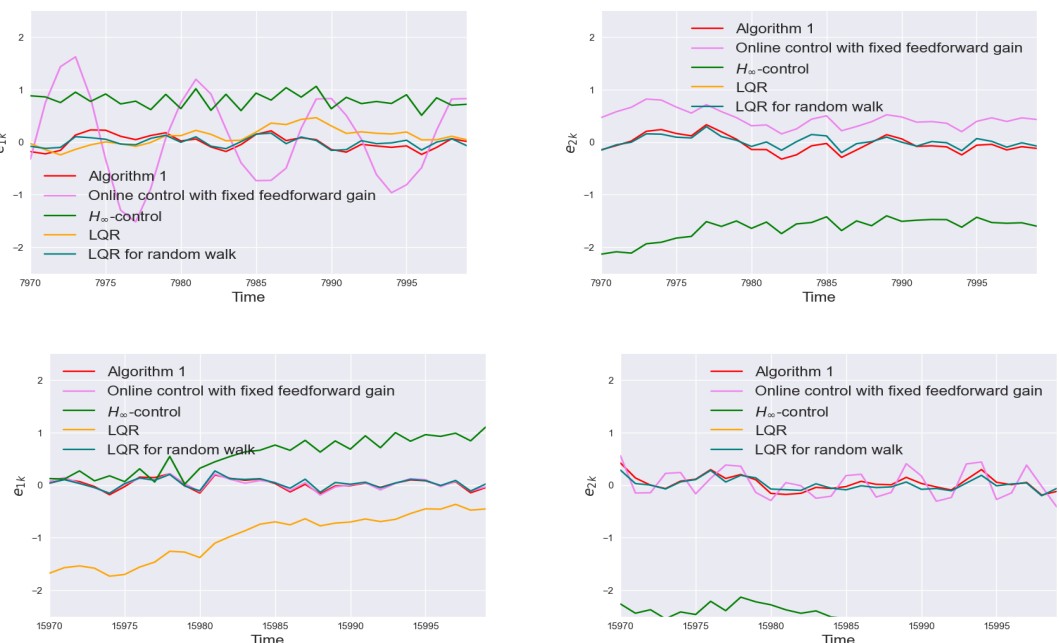

Figure 12: Tracking error over $T_{\text{change}} - 30$ to $T_{\text{change}}$ and $T - 30$ to $T$ for the random walk disturbance for the presented Algorithm 1, versus the online control algorithm with a fixed feedforward gain, the $H_\infty$ control, the LQR control, and the LQR for random walk using the reference signals in Fig. 10.

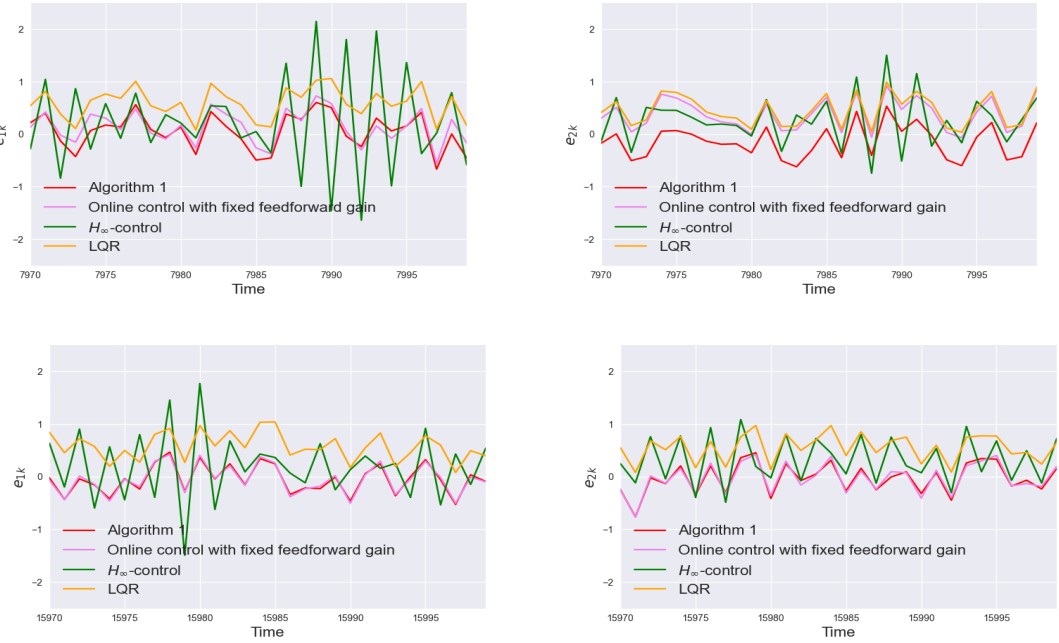

Figure 13: Tracking error over $T_{\text{change}} - 30$ to $T_{\text{change}}$ and $T - 30$ to $T$ for the uniformly sampled disturbance for the presented Algorithm 1, versus the online control algorithm with a fixed feedforward gain, the $H_\infty$ control and the LQR control using the reference signals in Fig. 10.

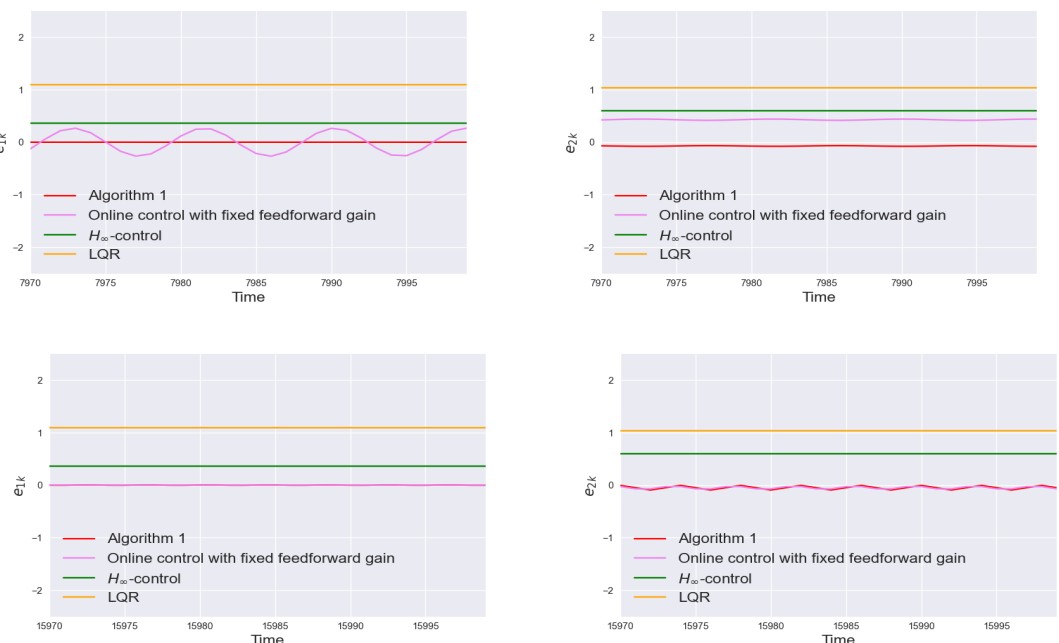

Figure 14: Tracking error over $T_{\text{change}} - 30$ to $T_{\text{change}}$ and $T - 30$ to $T$ for the constant disturbance for the presented Algorithm 1, versus the online control algorithm with a fixed feedforward gain, the $H_\infty$ control and the LQR control using the reference signals in Fig. 10.

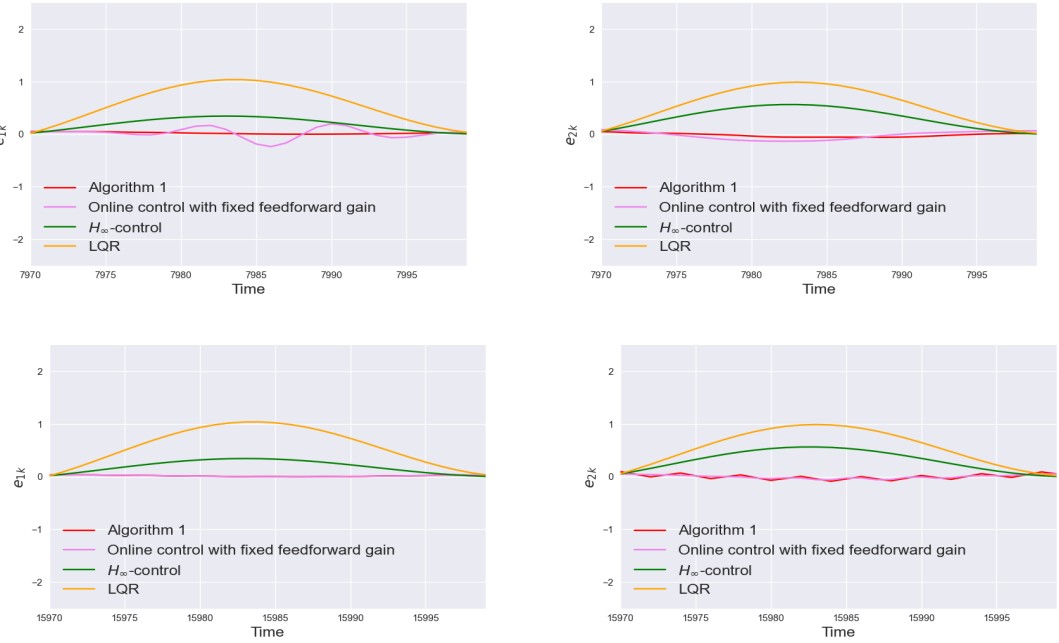

Figure 15: Tracking error over $T_{\text{change}} - 30$ to $T_{\text{change}}$ and $T - 30$ to $T$ for the amplitude modulation disturbance for the presented Algorithm 1, versus the online control algorithm with a fixed feedforward gain, the $H_\infty$ control and the LQR control using the reference signals in Fig. 10.

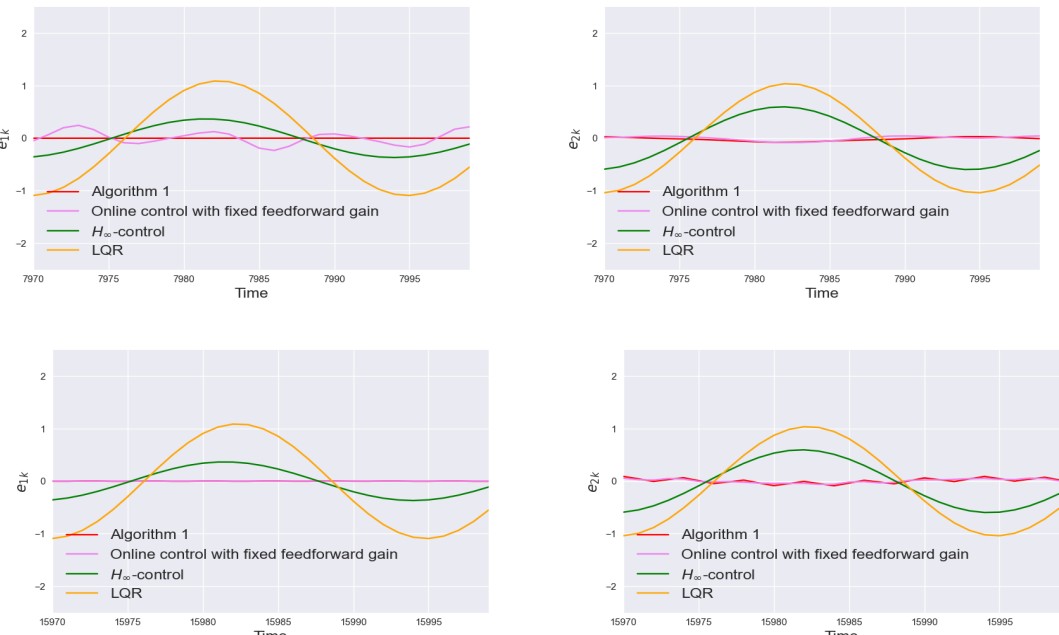

Figure 16: Tracking error over $T_{\text{change}} - 30$ to $T_{\text{change}}$ and $T - 30$ to $T$ for the sinusoidal disturbance for the presented Algorithm 1, versus the online control algorithm with a fixed feedforward gain, the $H_\infty$ control and the LQR control using the reference signals in Fig. 10.

