# OpenReview forum: "Online Optimal Tracking of Linear Systems with Adversarial Disturbances"
_TMLR — Accepted by TMLR_

### Review · Reviewer_RbB5 · 2022-12-29

**Summary Of Contributions:**

The authors propose to tackle the problem of optimal trajectory tracking for known linear systems with bounded additive disturbances.
The authors propose a method based on memory of past disturbances and values of the reference trajectory.

**Audience:**

No

**Broader Impact Concerns:**

None.

**Claims And Evidence:**

No

**Requested Changes:**

- Improve the list of references of existing methods that can solve the problem at hand, e.g., ADP methods, see Professor's Kyriakos G. Vamvoudakis work, adaptive methods including adaptive output regulators.

- Explain why the problem of output tracking for known linear models is more complex/useful than the problem of output tracking for unknown linear models ? I mean the later is solved by ADP methods (see again Vamvoudakis' work) and includes the simpler case of known linear models with unknown bounded additive uncertainties (which can be considered as part of the unknown linear model state matrix).

- Why citing Khalil's book for robust linear control ? This is not the proper reference for this control theory area; I suggest, Landau's books or J.C.Doyle's book, etc. Khalil's is the reference for deterministic nonlinear control.

- Speaking of Khalil's book, there you can find the so called 'Lyapunov reconstruction' for nonlinear models, which easily applies to linear models, and can solve your simple tracking problem for bounded additive uncertainties, if the upper bound is known, which is the case here.

- Notations need improvement: sometimes you use 't' and then use 'k' to denote times instances.

- You assume that the reference model for the trajectory is unknown, yet you use the observability index 'l' in the controller design, .e.g, equations 9, 19. How do you plan to compute 'l' ?

- Why is LQR the best controller for the case of Gaussian additive disturbances ? what about LQG ?! which is exactly meant for the case of Gaussian additive uncertainties.

- In the numerical section, and frankly all over the paper, you should compare your algorithm to other methods, e.g., adaptive control methods (e.g., adaptive output regulators), LQG, Lyapunov reconstruction, ADPs, etc. LQR/H_infty are not the best methods to use in this context.

- The analysis of the transient stability seems to be missing. The stability of the controller during learning is important if you want to use your method online, which seems to be the case here. Some boundedness guarantees between the disturbance bound and the tracking error bound need to be computed, e.g., Input-state stability arguments.



**Strengths And Weaknesses:**

Strengths:
- Clear presentation in terms of language
- Clear statement of the mathematical problem.

Weaknesses:

- The control problem is not as difficult or open as the authors are trying to convey
- There are several technical issues, i.e., strong assumptions, missing analysis, weak comparisons to existing methods, etc.
- Weak validation example.

---

> ### Author Response · Authors · 2023-02-02
> **Recommended changes.**
>
> References: We have now added citations from the work of Professor Kyriakos G. Vamvoudakis and other researchers in optimal tracking and output regulation parts.
>
> Comparison with ADP: We have now clarified that even though RL approaches based on the output regulation theory can deal with both the optimal tracking problem and disturbance rejection, the disturbance is assumed to be generated by a dynamical system. This, however, is rarely the case in most real-world applications, which limits the applicability of the output regulation theory. As can be seen from our simulation results, the standard feedback controllers that are also used and learned by ADP algorithms result in poor performance compared to the presented algorithm in case of non-Gaussian and general disturbances.
>
> Robust book and Lyapunov reconstruction: We have now cited J.C. Doyle's book in place of robust control theory. Besides, we have added Remark 4 to clarify that even though other control techniques, such as Lyapunov reconstruction and sliding mode control can be leveraged to design robust tracking controllers, they typically only concern the stability of the system. Therefore, they do not solve Problem 1 since no performance guarantees are provided.
>
> Notation: fixed
>
> Observability index: We do not need to know $l$ to design the controller. Indeed one can select $m_r$ large enough. In Theorem 4, we specify that $m_w= m_r=H$ and $H=\mathcal{O}(log \, T)$. We have added this point to Remark 3.
>
> LQR and LQG: LQR and LQG are sometimes used interchangeably when the system's states are assumed measurable. In the LQG setup, it is usually assumed that the system's state is not entirely measurable and only noisy measurements of system's output are available. A Kalman filter is then designed along with a linear controller. In the LQR setup, it is assumed that the system's states are measurable and there is no measurement noise. Since the states are assumed measurable in this paper, we used the terminology of LQR instead of LQG. We have now clarified this in the revised paper.
>
> Numerical section comparison: We have clarified that for systems with measurable states, LQR is actually the ground truth for comparison for Gaussian disturbance using full knowledge of the dynamics of the system and the reference generator. As we have now clarified, existing RL and ADP algorithms converge at best to the solution of the LQR and thus cannot guarantee good performance for the case where the disturbance is non-Gaussian. This is clearly shown in the simulation results section. Moreover, it is clarified that the Lyapunov-based design methods, such as Lyapunov reconstruction and sliding mode control in Khalil's book only guarantee boundedness and do not concern optimality. Therefore, they do not solve Problem 1.
>
> Transient stability: We have proved in Lemma 5 that the state variable is bounded, see Equation (35).  Following your suggestion, we have now given the bound for the tracking error in Lemma 6. We have also added Remark 9 to show that the average cost; i.e. $\frac{1}{T}\sum_{k=1}^{T}c_{k}(e_{k}, u_{k})$ remains bounded during learning.

---

> > ### Comment · Reviewer_RbB5 · 2023-02-21
> > **Thank you for your reply.**
> >
> > First, I want to thank the authors for their responses.
> >
> > Regarding the ADP being based on the output regulation theory, I do not agree, ADP is not based on the existence of a reference model, as it is the case in the output regulation, so I do not understand why the authors lumped the two together in their response.
> > Most importantly, ADP results do not assume the knowledge of A, B, which is their main advantage. Furthermore, you can design a nonlinear feedback using ADP, so your argument that you are testing against LQR and hence you are testing against the best ADP target gains is not correct.
> >
> > Regarding the Lyapunov methods, and more specifically the reconstruction method, being a regulation method only, I also do not agree, if you review some nonlinear control textbook (the Khalil might have a chapter about it, but you can also check the KKK book: Krstic, Kanellakopoulos and Kokotovi´c, Nonlinear and Adaptive Control Design, Wiley, 1995), you will see that a Lyapunov-based design has an underlying optimal cost, by inverse optimality results, and so by solving for a Lyapunov reconstruction controller you are designing an optimal controller for some given cost, i.e., in other words, you can tune the gains of the Lyapunov-based controller to make the tracking optimal and probably beat your optimal controller, since Lyapunov reconstruction leads to a nonlinear controller.
> >
> > Regarding LQR vs LQG, thanks for the review, but I am well aware of the differences. Yet, I am saying you can assume that your output is constituted of all the NOISY state vector and still apply LQG (with the noise filtering part) and not LQR to test against your controller in the case of Gaussian measurement noises.
> >
> > Thanks for adding Lemma 6, it does provide a bound and a safety guarantee during learning.

---

> > > ### Author Response · Authors · 2023-02-25
> > > **Response to Reviewer RbB5 comments**
> > >
> > > Thank you for your constructive comments and discussion to improve our paper. Please see our responses to your comments below:
> > >
> > >  ADP: We have now clarified in the revised paper that ADP methods optimize a risk-neutral (expected) or risk-aware measure of the cost function under the assumption that the noise is at least i.i.d and mostly Gaussian. This is because either the value function is directly learned (policy interaction or value iteration methods) based on collected data to estimate the expected or risk-aware cumulated rewards, or the expected or risk-aware cost function or its derivative with respect to the control parameters is learned directly using data (policy gradient methods). We are not aware of any ADP methods presented for adversarial disturbances. We have worked extensively on ADP before and believe it has excellent capabilities in dealing with uncertainties. However, new ADP developments are required when it comes to adversarial disturbances. We believe that the presented online learning approach can complement ADP, and their proper combination can bring the best of both worlds. Please also note that providing theoretical regret guarantees for nonlinear controllers is challenging and rare and can be considered as a future work.
> > >
> > > Lyapunov methods: We agree with reviewer that nonlinear controllers resulting from Lyapunov reconstruction and Sontag methods, which leads to inverse optimality, can result in high-performance controllers. We have now clarified in the simulation results that we only compared our results with linear control classes and cannot claim that it outperforms nonlinear control classes. Please note that no regret guarantees can be provided for inverse optimal methods since they guarantee optimality with respect to a meaningful cost, and not the pre-defined cost from which the regret is found.
> > >
> > > LQG: Please note that assuming the state measurement is noisy, our proposed algorithm is not applicable anymore. If the state variable is not exactly measurable, one cannot compute $w_{k-1}$. Note that measurable state variable is a common setup in control community. Extension to noisy state variable and more generally, partially observable and noisy system is indeed an interesting research topic and is the direction of our future works.

---

### Review · Reviewer_eMQ1 · 2023-01-12

**Summary Of Contributions:**

This paper formulates a tracking version of online linear control with convex costs. A variation of disturbance-based controllers with extra terms for tracking is proposed. It is shown that low-regret can be achieved by a variation of online gradient descent that is fairly similar to previous approaches to online control.

**Audience:**

Yes

**Broader Impact Concerns:**

None.

**Claims And Evidence:**

Yes

**Requested Changes:**

I would suggest the authors examine whether the work required to do tracking for this setup is really required or a simpler exact feedforward controller would suffice.

**Strengths And Weaknesses:**

Strengths:
The results appear to be rigorous. It is a novel problem that is of potential importance.

Weaknesses:
The biggest weakness is that the restrictive form of the tracking signal appears to enable a simple solution that is not pursued by the authors.

In particular, it is assumed that the reference signal is $r_k$ where
\begin{align*}
z_{k+1}&=Sz_k, \\\\
r_k &=Fz_k
\end{align*}
It is assumed that $z_k$ is not directly measured, and $S$ and $F$ are unknown.

The issue here is that in this setup, a valid realization for $(S,F)$ can be computed exactly in a finite number of steps. (I think the required number is $O(\ell)$, where $\ell$ is the observability index, but I haven't checked this part closely.)

In particular, take any $d\ge \ell$. Then for all $k\ge d$, $r_k$ satisfies a recursion of the form:
$$
r_k = -\sum_{i=1}^d \Theta_i r_{k-i}
$$
In this case, the observer canonical form given by:
\begin{align*}
S &= \begin{bmatrix}
-\Theta_1 & I & 0 & \cdots & 0\\\\
-\Theta_2 & 0 & I & \cdots  & 0\\\\
\vdots & & & \ddots & \\\\
-\Theta_{d-1} & 0 & 0 & \cdots & I\\\\
-\Theta_d & 0 & 0 & \cdots & 0
\end{bmatrix} \\\\
F &= \begin{bmatrix}I & 0 \cdots & 0\end{bmatrix}
\end{align*}
gives an observable realization of the process $r_k$. Furthermore, the initial condition, $z_0$, can be found by inverting the corresponding  observability matrix.

So, to get all of the data you need to completely predict all of the $r_k$ values, you just need to estimate $\Theta_i$ values. These can be computed by solving a system of linear equations.

At this point, the tracking part can be solved by just designing a feed-forward term, separately from the online learning component.

---

> ### Author Response · Authors · 2023-02-02
> **Reference Trajectory**
>
> We have now clarified in the new Remark 5 that it is possible to learn an observable canonical realization of the reference generator using measured data. Then, for the case of the linear feedback policy (3), the knowledge of the system model along with the learned model of the reference dynamics generators can be leveraged to 1) find the optimal feedback gain $K_{fb}$ by solving the algebraic Riccati equations, and 2) find the output regulator solutions $\Pi$ and $\Gamma$ by solving (5), and 3) find the feedforward gain $k_{ff}$ based on (4). However, to adapt to changes in the dynamics of the reference generator dynamics in adaptive control settings, offline learning methods are not satisfactory, since one has to compute the new estimate from scratch after a new sample becomes available. A popular approach for online learning for streaming settings is the gradient descent method. Nevertheless, as shown in Lemma 2, the optimization problem in hand is not convex in the gains of the linear feedback policy, which makes it intractable for online learning. Besides, one cannot find the optimal linear feedback policy for general adversarial disturbances offline, and the linear feedback policy, derived by solving the algebraic Riccati equation, performs poorly under general adversarial disturbances. Therefore, we leverage the new memory-augmented control policy (12) and guarantee a tight regret under general adversarial inputs. In this setting, even if the reference generator dynamics are learned using data, it is not clear how to find the feedforward gain since it also depends on the unknown gain $M$.

---

> > ### Comment · Reviewer_eMQ1 · 2023-02-17
> > **Issues are not resolved by revision**
> >
> > In my original review, I argued three main things:
> > 1) The reference dynamics can be learned exactly.
> > 2) After a finite number of steps, the entire future of the reference signal can be predicted (Because the initial condition can also be learned)
> > 3) A feedforward control term can be designed separately from the online learning component.
> >
> > The authors did a good job rebutting point 3). A naively designed feedforward controller done separately and offline would not be adequate for the task of online learning.
> >
> > However, the authors did not address the wider implications of how points 1) and 2) fundamentally change the problem. I will explain how they enable us to completely eliminate the effect of the reference signal, and reduce to the original problem from Agarwal et al.
> >
> > Point 1) in my review is analogous to Remark 1 of the paper: Even if you didn't know the dynamics of the disturbance at the start, you could quickly learn the model. So, in the paper, you use this assumption to explain why it is reasonable to assume we know A and B, since they can be computed to high accuracy (but not exactly). But you do NOT assume we know S and F, even though you can learn them EXACTLY!
> >
> > Now, Point 1) combined with Point 2), we actually know $z_k$ for all $k\ge 0$. As a result, of Point 1), $K_{ff}$ and $K_{fb}=\Gamma - K_{fb}F$ can found as in Theorem 1.
> >
> > Now, if we use $u_{lin} = K_{fb}x_k+K_{ff}z_k$ and change  coordinates to
> > $$
> > \begin{bmatrix}
> > e_k\\\\
> > z_k
> > \end{bmatrix}
> > $$
> >
> > where $e_k = x_k-r_k$, the property that $FS=AF+B\Gamma$ implies that the dynamics of (1) and (2) can be expressed as:
> >
> > $$
> > \begin{bmatrix}
> > e_{k+1} \\\\
> > z_{k+1}
> > \end{bmatrix} =
> > \begin{bmatrix}
> > A+BK_{fb} & 0 \\\\
> > 0 &  S
> > \end{bmatrix}
> > \begin{bmatrix}
> > e_k \\\\
> > z_l
> > \end{bmatrix}
> > +\begin{bmatrix}
> > w_k \\\\
> > 0
> > \end{bmatrix}
> > $$
> >
> > In other words, the dynamics of $e_k$ and $z_k$ are now completely decoupled.
> >
> > Now, if we consider, instead, memory augmented control policies of the form:
> > $$
> > u_k = K_{fb}x_k + K_{ff}z_k + \sum_{t=1}^{m_w} M^{[t-1]}w_{k-t}
> > $$
> > we get error dynamics of the form:
> > $$
> > e_{k+1}=(A+BK_{fb})e_k + w_k + B\sum_{t=1}^{m_w} M^{[t-1]}w_{k-t}
> > $$
> >
> > Now the design of the $M^{[t-1]}$ terms can be performed exactly as in the work of Agarwal et al.

---

> > > ### Comment · Reviewer_yjL7 · 2023-02-17
> > > **Some thoughts**
> > >
> > > I would like to add some comments here.
> > > Personally, the arguments that the authors make about the cases where reference generator dynamics change over time make sense and then there should be some algorithm that works in fully online fashion.
> > > However, as the reviewer points out, the current way of presenting the main theories may not be convincing enough to show its novelty in terms of performance guarantees because the generator dynamics are anyway assumed fixed although the algorithm itself may work "well" for adaptive scenario using its convexity (the algorithm itself is still beneficial in some cases).
> > > So, it would be great if there is any simple theoretical claim about the cases where the reference generator dynamics may be changing under certain constraints (will memory window size matter here?)
> > > Or is there any existing work that can be combined for those adaptive cases?

---

> > > ### Author Response · Authors · 2023-02-25
> > > **Reply to Issues are not resolved by revision**
> > >
> > > We thank both reviewers for the clarification. We believe that this comment helped us to better clarify the advantages of our online learning approach. We have now added a new solution and compared our results for based on the reviewer's comment. We call the proposed solution by the reviewer "The online control algorithm with a fixed feedforward gain" in the paper. We have added a remark and discussion to shown that the online control algorithm with a fixed feedforward gain might not return an optimal solution. Let $K_{fb}$ $^*$ denote the optimal linear feedback gain for the problem setup. Then, the associated feedforward gain, according to the tracking theory in Theorem 1, is $K_{ff}$ $^*$ $=\Gamma-K_{fb}$ $^* F$. In the online control algorithm, the parameter $M$ is learned such that $K x_k +\sum_{t=1}^{m_w} M^{[t-1]}w_{k-t}$ in (28) approximates $K_{fb}$ $^*$ $x_k$, see Lemma 5.2 of Agarwal2019 or Theorem 2 in this paper. However, the feedforward gain in the online control algorithm with a fixed feedforward gain is $K_{ff}=\Gamma-K F$ which is clearly different from $K_{ff}$ $^*$. It means that while the optimal linear feedback in the online control algorithm with a fixed feedforward gain is learned, the associated feedforward does not change. This may result in a higher cost, specifically when $K$ is away from $K_{fb}$ $^*$. Our simulation results clearly confirm this claim.
> > >
> > > We have also added some simulation results showing that the presented approach also works for time-varying reference dynamics. However, if batch learning approaches are used, the change in the dynamics of the reference must be detected to adapt to the new situation.

---

> > > > ### Author Response · Authors · 2023-02-25
> > > > **Response to some thoughts**
> > > >
> > > >  Please read our response above, where we have discussed that the proposed algorithm by the reviewer might not return an optimal solution. We have shown this in our simulation results also. So, even if the dynamics of the reference generator remains unchanged, our algorithm performs better.  Our proof of the regret bound is independent of the dynamics of the reference signal generator S and F. We make Assumption 3, stating that the reference signal is bounded and the output of the reference signal is measurable. We also assume that the classical state-tracking problem is solvable, such that the problem setup is meaningful. Since the information of S and F is not used, we expect that our algorithm works even if S and F change at some steps. Theoretical guarantees for changing  S and F however requires careful examination of the theories to define the minimum time intervals between dynamics changes and it is out of the scope of this paper. We have added simulation results for changing reference generator dynamics in Appendix H.

---

> > > > > ### Comment · Reviewer_yjL7 · 2023-02-27
> > > > > **Thank you for the response**
> > > > >
> > > > > Thank you for the clarifications.
> > > > > Just wanted to acknowledge that I read the response.
> > > > > Ok, so because Theorem 1 is under w=0 cases, if you design the "optimal" fixed gain after learning S and F may not be optimal under the authors' problem setups; this argument makes sense.
> > > > > The authors may want to refine Remark 10 a bit so that this point becomes clear; since this is one of the novelty aspects.
> > > > > (For example, why the authors' algorithm does not converge towards the one with fixed gain in general should be clarified more)
> > > > > It would be the case there is an another simple approach to design an optimal tracking controller if S and F are exactly known;
> > > > > I will add comments if I find that I am missing some points, but so far, it seems solid to me.
> > > > >
> > > > > I hope, based on this work and insights, there will be more practical variants coming out.

---

### Review · Reviewer_yjL7 · 2023-01-22

**Summary Of Contributions:**

This work presents an extension of Agarwal et al 2019 (Online control with adversarial disturbances) to an online reference tracking problems.
The reference signals are generated by an unknown but observable linear system.
It merges the theorem from Isidori 1989 (Theorem 1) and Agarwal et al 2019.

**Audience:**

Yes

**Broader Impact Concerns:**

No immediate concerns.

**Claims And Evidence:**

Yes

**Requested Changes:**

Major parts:
1. Assumption 3 the third one;  is this for all k>=0? if this is for all k, the boundedness may be a strong assumption indeed.
2. Remark 2; --- have arbitrary linear dynamics---  it is not arbitrary isn't it?  this kind of statement seems a bit misleading.
3. Can you mention the relation of this memory augmented controller to PI controller and disturbance observer?
4. eq 13 includes typos
5. Agarwal work is referenced, but please define truncated state, control and cost here.  Especially, it is hard to understand what the truncated cost is only from eq 18.  And equation 32 is referenced in page 7 but it is in Appendix and is hard to read.
This is true for the place like page 9 --- cost is computed according to equation 16-equation 18; how does one compute this cost?
6. Algorithm 1 also includes typo;  k_k+1 -> x_k+1?
7. Remark5 --- any convex c -> is it really "any"?  this is again a bit misleading
8. Although natural, but the definition 3 should be justified by a few sentences added to the current manuscript; because this is the optimality notion.
9. We assumed that the disturbance is bounded; why can we use Gaussian noise here?  If the disturbance is Gaussian, then this work cannot be applied or can we?   also the writing 0.1N(0,1) is a bit strange to me.
10. Experimental sections 6 are not clearly written.   6.2.3 to 6.2.6 seem too short; please use other ways of presenting them.  Also what are w_1k and w_2k?
There are several sayings "No other controller can beat the LQR...".  It again may leads to misunderstandings.  These statements are under different assumptions of known parameters.  If we know the noise generating mechanisms that are correlated somehow to the Gaussian disturbances, we may be able to do better by predicting the next noise magnitudes.  Please avoid this a bit strong sayings.
6.4 includes may redundancies that are already mentioned in the previous sections and are hard to follow.  Especially the figures are not really well explained.  Captions are not enough.  What is "counterfactual cost"? this saying seems appear here first time.  What is exactly the average cost?  averaged over time or runs?
11. in conclusions.  T is strangely typed.  Also the conclusion section is separated by figures and is hard to read.  Please deal with this.

12. Appendix D;   --- where the last inequality follows ----  -> the second last also follows from this?
Also in the proof of Lemma5  what is "||A||x"?  x should be in norm?  and the norm notation is missing for the same line for the last term; this makes a bit confusing to follow.   And this appendix does not change line when writing "<" or ">" symbol while the other parts of appendices do so; please make the writing consistent.  And in Lemma 6,  Ytilde has all its elements....  this Ytilde refers to Y_H,k or Y_t?  And the bottom of page 22  what is L_f3G...?   Before equation 42, what do you mean "Therefore, in general, ..."?  What does this "general" mean for?
Finally, did you correctly define the norm || * ||?  is this spectral norm?
All of these make it harder to follow.  Even if they are immediate from the existing work like Agarwal et al., please carefully review the writings to increase the readability


Minor parts:
1. page 1  ---Gaussian nor generated by a dynamic system.  -> dynamical system?
2. It would be great to see a few sentences in the introduction to justify this reference tracking with adversarial noise problem in practice.
3. Page 3 in Regret ---(usually the optimal)--- -> this is talking about the notion of optimality and is presenting regret; so saying "optimal" here is confusing; what does this "optimal" means here?
4. page 3 --- parameterized as it usually is in control theory--- can you elaborate a bit on this?  what do you mean by parameterize controls?
5. page 3 notation ->  xi:j and xi,j are mixed.
6. page 5  ---- design a control policy pi: (x..  w 1:k ...)  -> (w1:k-1)?
7. page 7 circumstance -> circumvent?
8. Remark 5   worse case -> worst case?
9. page 10 what do you mean by "disturbance is discontinuous and noisy"?  it is a bit inaccurate description
10. page 11 top   Not -> note?
11. Appendix B   --- not convex in Kfb Kff.   -> perhaps add "in general"?

**Strengths And Weaknesses:**

Strength:
1. For real applications, reference tracking is very important; for example for quadrotor path tracking or leader follower problems; in that sense, this extension benefits practitioners to design optimal algorithms.

2. The flow of the paper is easy to follow; and some parts are well-written.

3. It does a good list of comparison experiments which concisely summarizes the motivations of this work

Weakness:
1. There are many technically confusing or inaccurate descriptions with several typos; (I will list hem in the requested change section); especially, since this work is largely inspired by Agrawal et al 2019, and is rather aiming for its practical benefits of extensions, the technical accuracy is very crucial.

2. Related to the above, some experimental settings seem contradicting to the assumptions; also, some parts seem written in rush, and are showing inconsistency of writings. (I will list them in the requested change section)

3. Experiment contents are good, but the way they are shown are a bit rough. (I will list them in the requested change section)

---

> ### Author Response · Authors · 2023-02-02
> **Major parts**
>
> 1- Assumption 3: We have assumed that the reference is bounded for all $k \geq 0$. We have clarified in Remark 2 that even though this assumption does not cover all types of references,  it can generate a large class of useful command trajectories, including unit step, sinusoidal waveforms, damped sinusoids, etc. Removing this assumption has always been a longstanding challenge in the optimal tracking literature and is the direction of our future works. For an unbounded reference signal, the average cost in (10) is not bounded.  The reason is that the controller in tracking contains a feedforward term from the reference signal, see (4), and if the reference is unbounded, then the average cost is unbounded. So, such an assumption is usually considered when studying average cost, see for example Abbasi-Yadkori et al. (2014), Adib Yaghmaie et al., 2019. It might be possible to have an infinite average cost but a finite regret; but the regret decomposition and analysis are intractable in this case. We would like to add that tracking unbounded reference signals is usually studied in the discounted cost settings where it is possible to guarantee the boundedness of the discounted cost by selecting the discounting factor properly, see for example, Kiumarsi et. al. (2014).
>
> 2- Arbitrary linear dynamics: Yes, you are right. Arbitrary is now removed.
>
> 3- PI controller: We have now clarified in the first paragraph of the introduction, as a special class of the internal model approach, a Proportional-Integrator (PI) controller contains an integrator so it can be used for tracking a constant reference signal or rejecting a constant disturbance. Our proposed controller in (12) belongs to the ''feedforward approach'' to solve the tracking problem. To be more specific, the controller in (12) contains a feedforward term from the reference signal and it does not contain an internal model of the reference signal. We have added this point to the third paragraph of the introduction where we introduce our proposed approach briefly.
>
> 4- Eq 13: The typo is fixed.
>
> 5- Truncated state and cost: Equation (32) has been moved to the body of Lemma 3 now. Remark 6  is added regarding the computation of the truncated state, input, and cost.
>
> 6 and 7- The typo in Algorithm 1 is fixed, and in the statement "any convex set," the word "any" is removed to avoid confusion.
>
> 8- Justification for Definition 3 is added right after the definition.
>
> 9- Gaussian noise: Even though our theoretical results require bounded disturbance, the bound does not need to be known and could be large. This is in contrast to the robust control methods such as $H_{\infty}$.  In our simulation results, we have considered Gaussian disturbance:  To the best of our knowledge, the Gaussian noise generator (numpy.random.normal) by Numpy in Python generates bounded samples. We have now used $\mathcal{N}(0,0.01)$ and clarified this point in the simulations.
>
> 10- Experimental sections: We have removed subsections 6.2.3-6.2.6 and listed all disturbances under Subsection 6.2. We have now mentioned that the disturbance in the simulation is of dimension two and $w_{k}=[w_{1k},w_{2k}]^T$. The statements like "No other controller can beat the LQR..." have been removed to avoid ambiguity. Section 6.4 has been further refined, and redundant explanations have been removed. The caption for Fig. 1 has been rewritten to be more informative. The average cost had already been defined in equation (10). We have now referred to this equation to increase readability. The term "counterfactual" has no longer been used in the paper.
>
> 11- $T$ is fixed in Conclusion.
>
> 12- Typos: 1- Appendix D: It was a typo. The second inequality follows from $(\kappa,\gamma)$. It has been fixed now.
> 2- Lemma 5: Typos have been fixed. The correct notations have been now used consistently throughout the paper.
> 3- Lemma 6: Fixed.
> 4- The number "3" in Equation (39) was misplaced and it has now been fixed.
> 5- Regarding "general" in the proof of Lemma 6: It refers to combining two inequalities. We have now numbered those inequalities and mentioned that by combining those two inequalities, we get the result.
> 6-  We have now added to the notations that for matrix $A$, the spectral norm is used.

---

> > ### Author Response · Authors · 2023-02-02
> > **Minor parts**
> >
> > 1- "dynamic system" is now changed to "dynamical system",
> >
> > 2-Introduction is revised to justify reference tracking with adversarial noise.
> >
> > 3- The term "optimal" is now removed.
> >
> > 4- More explanations have been added in this regard. Examples of parameterized controllers are state feedback or disturbance-action policy Agarwal et al. (2019) where the control input is parameterized based on the state or disturbance and the parameters are gains that are designed. Parameterization of the controller allows analysis in the context of control theory.
> >
> > 5, 6, 7, and 8- Fixed
> >
> > 9- We have removed this sentence and mentioned that the disturbance is randomly generated.
> >
> > 10 and 11- Fixed accordingly.

---

> > > ### Comment · Reviewer_yjL7 · 2023-02-03
> > > **Comments**
> > >
> > > Thank you for the revision; it looks much better now.
> > > I will add comments if I find any additional concerns later.

---

### Decision · Action_Editors · 2023-03-19

**Recommendation:** Accept with minor revision

**Comment:**

**Summary:**
The paper considers a linear dynamical system with a known dynamical system and provides an online algorithm for tracking a reference signal of an unknown linear dynamical system under an adversarial disturbance. The algorithm minimizes an additive cost function, which belongs to a subset of convex loss functions, and provides regret guarantee compared to the best linear feedback controller in hindsight.


**Evaluation:**
After discussions between the reviewers and the authors, and two revisions of the papers during the discussion period, the reviewers all agree that this paper can be accepted.


Although the reviewers believe that this paper's contribution may not be the most significant one in this line of work (eMQ1) or that the proposed method may not be practically as useful as other (nonlinear) controllers that can presumably perform better (RbB5) (though without any regret guarantees, as this paper), it still has notable contributions and provides reasonably good evidence to support it that warrant its publication.

After reading the revised version of the paper myself and comparing notes with the reviewers, I **recommend the acceptance of the paper with minor revisions.**

The minor revision is requested to ensure that the paper goes through a careful round of revising by the authors and be as clear as possible. This will hopefully make it a better and more readable paper.


**Comments:**



*The writing of the paper has room for improvement.*
Some examples:
- Some of the remarks are clearly reactions to the reviewers' comments. They might be adequate for the reviewers, but they disrupt the flow of the paper for a reader who is not aware of the discussions. Remark 5 is an instance. Please revise the paper by making these remarks a more integrated part of the paper.

- Different types of disturbances in Section 6.2 follow an odd order: the paper first describes the "fourth case" (Gaussian), then "fifth case" (Random walk), etc.

- There are a few typos in the paper. For example,
- [p1] "The account for the effect ..." should be "To account for ..."
- [p21] "... matrix can be spans by ..." should be "... can be spanned by ..."
Other reviewers have also spotted typos, so please fix all of them, if you have not already done so.


*I have some concerns about the statement of Theorem 4 and its consequences, and I believe it requires some clarifications:*

+ First, is it assumed that T is fixed? This looks like the case because the choice of learning rate and H are T-dependent (and not t-dependent). If that is the case, one cannot simply let T goes to infinity as in Remark 9. One can use a doubling trick to deal with it, but that should be stated and analyzed clearly.

+ Second, the last conclusion of Remark 9 that the LHS converges to the RHS is not correct. The LHS is less than or equal to RHS, but the regret guarantee does not necessarily show the convergence. Your algorithm might actually be better than the best linear feedback controller.


*Why is the regret defined based on the best linear feedback controller in hindsight?*
If I understand correctly, the linear feedback controller, defined in Eq. (3), is the right class when there is no disturbance, but it is not necessarily the right class when we have disturbances. Am I missing something?

*Assumption 5 is strong and the choices of $\Pi_P$ and $\Pi_M$ in the algorithm are unclear.*
The second part of Assumption requires M and P obtained by the algorithm be bounded. This sounds like a requirement on the algorithm and not an assumption. So, I wonder whether this boundedness is enforced by the choice of $\Pi_P$ and $\Pi_M$ in the algorithm? The paper is not clear about it.
If it is not enforced, why should we expect a gradient descent algorithm stay bounded?


**Audience:**

Yes. Although the work addresses a problem and extensively uses tools from the control theory, for which TMLR may not be the best venue, it uses tools from Online Learning. This makes the paper of interest to the sub-community of TMLR who work at the intersection of (online) learning and control theory.


**Claims And Evidence:**

Most of them. There are some minor issues that require further clarifications.

---

> ### Author Response · Authors · 2023-04-04
> **Reply to Action Editors Comments**
>
> Thank you very much for your thorough reading of our paper and your valuable comments. We have now revised the paper according to your comments and made the following changes.
>
> Remarks:  We have now incorporated two remarks (old Remarks 4 and 9) into the text and rewritten one remark (old Remark 5) to enhance readability.
>
> Typos:  Typos are now fixed. We have undertaken thorough proofreading to ensure there are no typos.
>
> $T$:  You are right. $T$ is fixed. We have added this point to Theory 4. We have removed $T\rightarrow \infty$ in the discussion after the theorem.
>
> Regret bound:  You are right. We have removed this part and only mentioned that the regret analysis gives an upper bound for the average cost by Algorithm 1.
>
> Regret:    The reason for selecting the class of linear policies as the baseline is twofold. Firstly, since we do not have any information about the disturbance, we compare it against the best policy when there is no disturbance in the system, which is the class of linear policies. Secondly, we select the class of linear policies so that the theoretical analysis of the regret is possible. This class has also been used by Agarwal et al. (2019) and others.
>
> Assumption 5: The bounds on $M,\: P$ are a prerequisite for proving the theoretical results related to our algorithm. It is enforced by using projected gradient descent in the algorithm.
> We have now clarified in the explanation of Algorithm 1 that we perform projected gradient descent and $\Pi_{M},\:\Pi_{P}$ denote projection onto the set of matrices with appropriate dimensions and bounded norms as specified in Assumption 5.

---

> > ### Comment · Action_Editors · 2023-04-15
> > **Thanks for the update!**
> >
> > Thank you for the revisions. The paper looks good. I have two further questions/remarks:
> >
> > 1) Is $\Pi - F = 0$ in the second part of Equation (5) correct, or is something missing there?
> > If it is correct, then the only unknown is the first equation in (5) is $\Gamma$, which is the solution of $B \Gamma = F S - A F$. Am I interpreting this correctly?
> >
> > 2) It might be better to provide an exact reference of Theorem 1 in (Isidori, 1989)? That is a long textbook, and it may be difficult to find that result without a more precise pointer.

---

> > > ### Author Response · Authors · 2023-04-17
> > > **Reply to Action Editors Comments-second round**
> > >
> > > Thank you very much again for your constructive comments to improve the paper. Here are our responses to your questions:
> > >
> > > 1- Please note that equation (5) is correct. If the dynamics of the reference generator are known or learned, $\Gamma$ is the only unknown variable. Note, however, that one cannot use $\Gamma$ to design the feedforward gain offline. The reason is that the optimal feedback gain $K_{fb}^*$ is unknown and the parameter $M$ is learned such that $K x_k +\sum_{t=1}^{m_w} M^{[t-1]}w_{k-t}$ approximates $K_{fb}^* x_k$. So, one cannot simply set $K_{ff}^*$=$\Gamma-K_{fb}^* F$. We have discussed this point in Remark 8.
> > >
> > > 2-The problem setup in (Isidori, 1989) is for continuous-time systems setup. We have now referred to Theorem 1.35 and Remark 1.36 of (Huang, 2004) for our Theorem 1, which is for discrete-time systems. Note that the results are the same for continuous and discrete-time systems.

---

> > > > ### Comment · Action_Editors · 2023-04-17
> > > > **Thanks!**
> > > >
> > > > Thanks for your clarifications.